# Oryx MLLM: On-Demand Spatial-Temporal Understanding at Arbitrary Resolution

**Zuyan Liu[1,2][*], Yuhao Dong[2,3][*], Ziwei Liu[3], Winston Hu[2], Jiwen Lu[1][†], Yongming Rao[2,1][†]**
[1] Tsinghua University   [2] Tencent   [3] S-Lab, NTU
{liuzuyan19,raoyongming95}@gmail.com

## Abstract

Visual data comes in various forms, ranging from small icons of just a few pixels to long videos spanning hours. Existing multi-modal LLMs usually standardize these diverse visual inputs to fixed-resolution images or patches for visual encoders and yield similar numbers of tokens for LLMs. This approach is non-optimal for multimodal understanding and inefficient for processing inputs with long and short visual contents. To solve the problem, we propose Oryx, a unified multimodal architecture for the spatial-temporal understanding of images, videos, and multi-view 3D scenes. Oryx offers an on-demand solution to seamlessly and efficiently process visual inputs with arbitrary spatial sizes and temporal lengths through two core innovations: 1) a pre-trained OryxViT model that can encode images at any resolution into LLM-friendly visual representations; 2) a dynamic compressor module that supports 1x to 16x compression on visual tokens by request. These designs enable Oryx to accommodate extremely long visual contexts, such as videos, with lower resolution and high compression while maintaining high recognition precision for tasks like document understanding with native resolution and no compression. Beyond the architectural improvements, enhanced data curation and specialized training on long-context retrieval and spatial-aware data help Oryx achieve strong capabilities in image, video, and 3D multimodal understanding.

## 1 Introduction

Multi-Modal Large Language Models (MLLMs) have made significant strides in processing and integrating visual and linguistic inputs to generate coherent and contextually relevant responses. Proprietary models such as (OpenAI, 2023b; 2024; GeminiTeam, 2024) exemplify the cutting-edge capabilities of MLLMs. Concurrently, the open-source community is actively advancing MLLMs by enhancing their ability to understand diverse visual content (Tong et al., 2024; Liu et al., 2024g; Yang et al., 2023a; Dong et al., 2024; Liu et al., 2025), including images (Li et al., 2024a; Chen et al., 2024b), videos (Lin et al., 2023a; Cheng et al., 2024; Qian et al., 2024), and 3D data (Hong et al., 2023), *etc*. As MLLMs become stronger, there is a growing need for more general and unified MLLMs that are capable of processing visual content in more diverse forms and accomplishing more challenging multimodal problems.

One core challenge in the path to achieving more general MLLMs is to develop better visual representations for diverse visual data. Visual data exhibit significant complexity and diversity, characterized by variations in collection sources, targeted visual tasks, specific contents, and resolution qualities. Existing approaches often simply treat all kinds of visual inputs uniformly, overlooking the variations in visual content and the specific demands of different applications. For example, early MLLMs (Alayrac et al., 2022; Li et al., 2023; Bai et al., 2023) attempt to standardize these diverse visual inputs by converting them into a fixed resolution so that pre-trained CLIP encoders can be used to extract high-quality visual representations that are well aligned with language contents. Recent advancements in MLLMs (Liu et al., 2024c; Xu et al., 2024b; Yao et al., 2024) extend the idea by introducing dynamic partitioning (Liu et al., 2024c) as a means to produce high-resolution visual representations while utilizing the strong CLIP models for encoding. However, the solution remains a compromise due to the lack of high-quality multi-modal encoders that support native resolution inputs.

---

[*]Authors contributed equally to this research. [†]Corresponding authors.

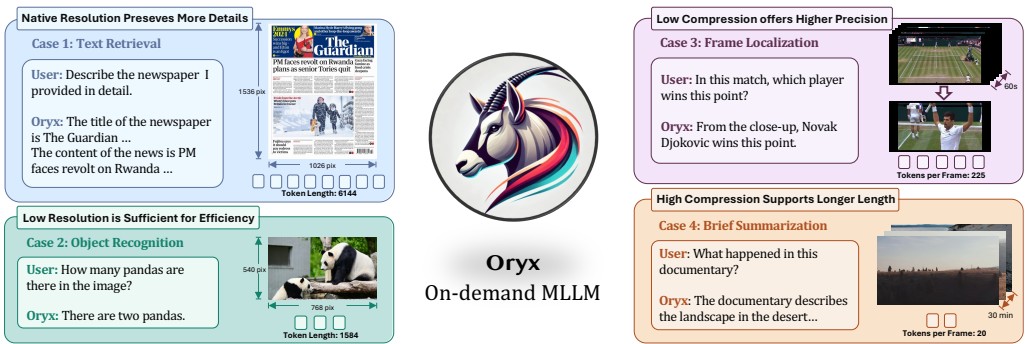

Figure 1: **Our main idea of on-demand multimodal understanding.** Different visual data and tasks may require different input resolutions and compression ratios on visual tokens. Supporting arbitrary resolution in an on-demand manner for visual inputs emerges as a more general and effective solution for visual understanding in MLLMs.

Supporting native resolution in an on-demand manner for visual inputs emerges as a more generalized and effective solution for visual understanding in MLLMs, offering several advantages: it prevents information loss by utilizing the entire image as input, thereby resolving extreme corner cases, and it enhances efficiency and naturalness, resulting in better overall performance. As illustrated in Figure 1, optimizing for resolution and compression can lead to greater efficiency and meet practical needs: high resolution is crucial for text-relevant tasks, while object-level tasks may require only simple images, some applications may need to summarize extremely long videos while others maintain high precision for each frame.

In this paper, we explore on-demand MLLMs for comprehensive spatial-temporal understanding by introducing evolved architectural designs and propose the new Oryx model, which aims to address these challenges and enhance the functionality of MLLMs. Oryx is a unified spatial-temporal understanding MLLM framework that adeptly handles arbitrary visual resolutions, varying temporal lengths, and a diverse range of tasks in an on-demand manner. Oryx is characterized by the following key contribution: 1) A pre-trained visual encoder OryxViT is developed to generate LLM-friendly visual representations at native resolutions. Equipped with adaptive positional embeddings and variable-length self-attention, OryxViT can efficiently process visual data with different sizes in parallel; 2) Dynamic compression technique that adjusts downsampling ratios arbitrarily while fusing the information through a shared projector, thereby supporting a seamless switch between 1x to 16x compression. The new design enables Oryx to easily process extremely long inputs with up to 16x compression while maintaining high recognition precision for inputs that do not require compression; 3) Enhanced data curation and training strategies that help Oryx achieve pioneering performance in multimodal images, videos, and 3D data understanding and easily adapt to arbitrary input resolution and tasks simultaneously.

We evaluate the Oryx model on a wide range of multi-modal benchmarks, demonstrating remarkable performance in both spatial and temporal understanding across image, video, and multi-view 3D data. Notably, the Oryx model excels in general and long-form video comprehension, achieving competitive results with a 7B model size and surpassing models up to 72B in size with our 32B variant. This has led to new state-of-the-art results among open-source models on several benchmarks, including NextQA (Xiao et al., 2021), Perception Test (Patraucean et al., 2024), MMBench-Video (Fang et al., 2024), and MVBench (Li et al., 2024c) for general video understanding and MLVU (Zhou et al., 2024), LongVideoBench (Wu et al., 2024) for long-form video benchmark. Additionally, the Oryx model shows strong performance in 2D and 3D spatial understanding, outperforming mainstream image-based MLLMs and 3D-specific LLMs, benefiting from its unified training strategy.

## 2    RELATED WORK

**Visual Encoding in Multi-Modal LLMs.** Multi-modal LLMs depend on visual encoders to extract visual features and employ connectors for aligning visual features with the LLMs. Alayrac et al.

(2022) and Li et al. (2023) utilize attention to capture visual features and align the visual encoder with LLMs through learnable queries, which may struggle when not adequately trained. LLaVA (Liu et al., 2024d;b;f) utilizes a simple MLP to connect the visual encoder with LLMs, while Ranzinger et al. (2024) combines visual features from different encoders for enhancement. However, they are limited to fixed resolutions, which may hinder their ability to capture detailed information and restrict their flexibility in understanding images with varying aspect ratios. Recent advancements in high-resolution perception (Liu et al., 2024c; Xu et al., 2024b; Yao et al., 2024) have primarily been driven by dynamic partitioning, which divides an image into multiple patches of equal resolution. While this method can manage high-resolution images, it is inefficient, and the partitioning process may result in the loss of critical information present in the original image. In this paper, we introduce OryxViT, an innovative step in visual encoding that enables native resolution perception.

**Multi-modal LLMs Supporting Diverse Contexts and Tasks.** Recent advancements in MLLMs have enabled them to comprehend a wide range of complex visual inputs from different tasks with various contexts. Lin et al. (2023a); Cheng et al. (2024); Qian et al. (2024) try to combine image and video perception, and Zhang et al. (2024a) focuses on long-form video analysis with extended context lengths. 3D-LLM (Hong et al., 2023) made the first attempt to enable MLLMs to comprehend 3D environments. Li et al. (2024b); Jiang et al. (2024) investigate interleaved data training to handle multi-image scenarios, and Li et al. (2024a) unifies single-image, multi-image, and video settings through improved data curation and training strategies. While previous approaches relied heavily on enhanced data curation to achieve multi-task comprehension, we propose a novel framework that represents complex visual inputs with cohesive representations. Our model is capable of processing visual contexts of arbitrary sizes, videos of varying lengths, and 3D data seamlessly, supporting various context lengths and versatile tasks.

## 3 METHODS

In this section, we provide a detailed explanation of Oryx's contribution. Our design is segmented into two primary components: the architecture and the data curation & training pipeline, which are elaborated upon in Section 3.1 and 3.2, respectively. We describe our innovative architecture to process native and on-demand visual inputs within MLLMs, as illustrated in Figure 2, enabling the development of a model capable of generalizing across image, video, and 3D data. Furthermore, we outline the simple yet effective training pipeline of the Oryx model.

### 3.1 ORYX ARCHITECTURE: MLLM WITH NATIVE AND FLEXIBLE VISUAL INPUTS

#### 3.1.1 VISUAL REPRESENTATIONS WITH NATIVE RESOLUTION

Resizing and regularizing visual inputs, including images and videos, is a necessary and effective preprocessing step. Common practice typically involves resizing and cropping visual inputs to a fixed resolution with a square shape. However, such processes may negatively impact the performance of vision backbones, as previous studies on vision recognition have demonstrated the effectiveness of maintaining visual content in its original form. NaViT (Dehghani et al., 2024) leverages the characteristics of the vanilla ViT (Dosovitskiy, 2020), introducing a pack sequence operation that accommodates images of any aspect ratio and resolution for efficient training. Similarly, Flex-iViT (Beyer et al., 2023) and ViTAR (Fan et al., 2024) incorporate randomly resized images during training to develop a Vision Transformer capable of handling inputs of varying resolutions.

Despite these advancements, the effectiveness of native or arbitrary resolution in the realm of MLLM has barely been explored. Most existing MLLMs integrate original image-text visual encoders such as CLIP (Radford et al., 2021) and SigLIP (Zhai et al., 2023) to encode input visual data. We posit that MLLMs provide an optimal environment for processing visual representations at their native resolution for two primary reasons: (1) the sources and tasks associated with visual inputs are diverse, necessitating varying demands and formats; (2) the token lengths in MLLMs are inherently dynamic, particularly in the language component. Consequently, the dynamic representation of visual context aligns seamlessly with subsequent processing stages.

In Vision Transformer (ViT) models (we omit the class token here for simplification), given the visual input $\{x\}^{\in H \times W}$, where typically $H \neq W$, the ViT first resizes the visual input into $\{x\}^{\in N \times N}$.

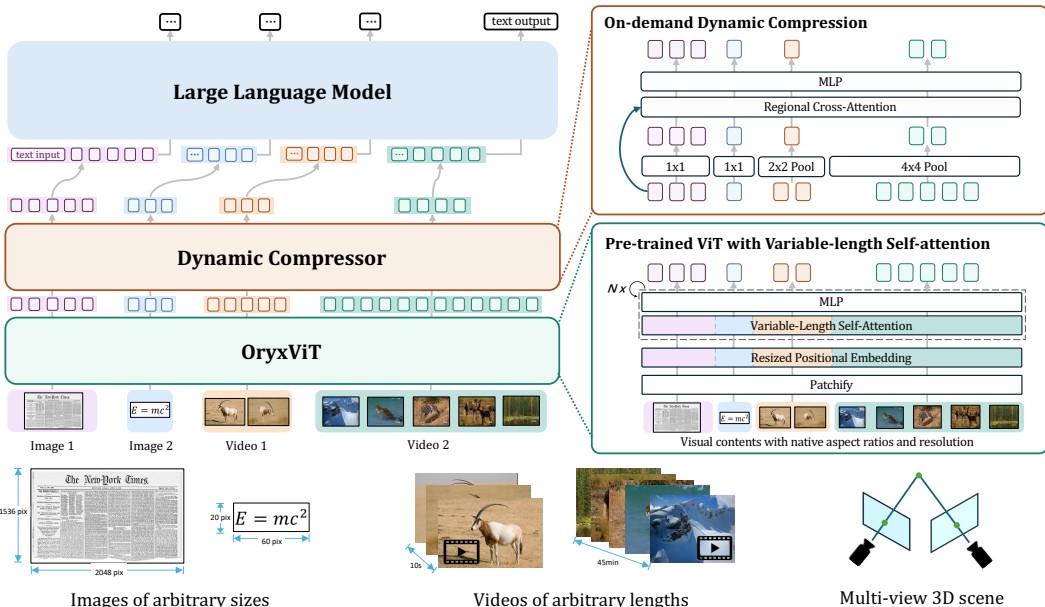

Figure 2: **Overview of Oryx architecture.** Oryx offers two options to process visual inputs with arbitrary spatial sizes and temporal lengths in an on-demand manner. 1) A pre-trained OryxViT equipped with variable-length self-attention to encode visual features with native aspect ratios and resolution. 2) A dynamic compressor offering on-demand compression on visual tokens while maintaining a unified token form.

The resized image is then passed through patch embedding layers, which partition the image into patches of size $p \times p$, resulting in a sequence of patches $\{x\}^{\in (N/p) \times (N/p)}$. Conventional Vision Transformers utilize a fixed-size position embedding matrix $P$ corresponding to the predefined image size $N \times N$. However, when processing visual inputs at their native resolution $\{x\}^{\in \lfloor H/p \rfloor \times \lfloor W/p \rfloor}$, directly resizing $P$ to $\lfloor H/p \rfloor \times \lfloor W/p \rfloor$ can lead to a significant drop in accuracy, as demonstrated in previous works (Dehghani et al., 2024; Beyer et al., 2023).

To address the issue of native resolution processing, we introduce a visual encoder named OryxViT, which builds upon the advanced SigLIP (Zhai et al., 2023) models and is based on the Vision Transformer (Dosovitskiy, 2020) architecture. We modify the vision encoder by incorporating a sufficiently large position embedding matrix $P$ that accommodates the maximum target input sizes. For each visual input, we rescale the original position embeddings into $P^{\in \lfloor H/p \rfloor \times \lfloor W/p \rfloor}$ using bilinear interpolation and apply the transformation $x = x + P$. The pre-training strategy for the proposed visual encoder OryxViT under native input resolution follows the training format of common MLLMs. We employ a relatively lightweight LLM as the language interface, keeping the vision encoder's parameters unfrozen while freezing most of the other parameters. Details for the training settings and datasets can be referred to in the Appendix.

A significant challenge is managing the dynamic sequence length $N = \lfloor H/p \rfloor \times \lfloor W/p \rfloor$ for the Vision Transformer during batch processing, where we propose the Variable-Length Self-Attention strategy to address this issue. For visual patches with lengths $N_1, N_2, \ldots, N_b$ in a batch of size $b$, we concatenate the patches across the sequence dimensions into a shape of $[1, \sum_{i=1}^{b} N_i, C]$ before feeding them into the transformer blocks. We utilize the variable-length attention operator provided in flash attention (Dao et al., 2022) to compute the attention for each visual input within the batch independently. With these designs, our OryxViT can efficiently process visual signals of varying aspect ratios in batch mode, maintaining a forward speed comparable to that of conventional fixed-resolution visual encoders. We also provide the efficiency test for OryxViT in the Appendix.

### 3.1.2 ON-DEMAND DYNAMIC COMPRESSION SUPPORTING LONG VISUAL CONTEXT

With visual inputs varying in temporal length and resolution, such as some video data lasting tens of minutes, treating all inputs equally, as in most previous works (Zhang et al., 2024a; Xue et al., 2024),

leads to inefficient computational costs. To address this, we propose a Dynamic Compressor, which is capable of performing higher compression ratios for longer contexts. Our design unifies visual contexts with different compression ratios into a consistent pattern, allowing us to control the overall visual sequence length on demand.

Using the visual representation feature map $f$, the compression serves as the bridge between vision and language modalities. We implement downsample layers with varying ratios to accommodate different input lengths. Specifically, we categorize the visual context into pure images, short videos, and long videos, applying downsample layers $d_1, d_2, d_3$ respectively. We satisfy the downsampling ratio $r_1 < r_2 < r_3$ for layer $d_1, d_2, d_3$ to reduce the token length for frames in videos.

We obtain the low-resolution feature map $f_L = d_i(f_H), i = 1, 2, 3$ from the high-resolution feature map $f_H$. To mitigate the effects of downsampling, we employ an attention operation to facilitate interaction between $f_L$ and $f_H$. Specifically, for a downsample ratio $r$, we treat $f_L \in \mathbb{R}^{N \times C}$ as the query tensor $\mathbf{Q}$ and $f_H \in \mathbb{R}^{N \times r^2 \times C}$ as the key tensor $\mathbf{K}$ and value tensor $\mathbf{V}$. Each patch in the low-resolution $f_L$ interacts with $r^2$ neighboring patches in the high-resolution $f_H$ through a cross-attention operation, formulated as follows:

$$f_L = f_L + \text{Softmax}(\frac{\phi_q(\mathbf{Q})\phi_k(\mathbf{K}^T)}{\sqrt{d_k}})\mathbf{V} \tag{1}$$

where we define the query and key projection layers, denoted as $\phi_q$ and $\phi_k$, to project the query and key tensors into lower dimensions. To maintain the original features from the visual encoder and limit the number of linear projection layers, we omit the value and output projection layers commonly used in attention modules. Then we utilize a shared MLP across multiple downsample modules to project the compressed low-resolution features into the embedding space of the language model. We preserve the interactions between different downsample ratios through the shared projection. Upon completion of the dynamic compression module, the final visual representation features are flattened and integrated into the sequence of visual tokens among the text tokens. This combined sequence is then fed into the language model for token prediction.

## 3.2 DATA CURATION & TRAINING PIPELINE

### 3.2.1 ONE MODEL FOR ALL: IMAGE, VIDEO, AND 3D UNDERSTANDING

Previous work (Li et al., 2024a; Chen et al., 2024b; QwenTeam, 2024b) has demonstrated the coexistence of MLLMs that support both image and video modalities. Building on this foundation, our research aims to extend the capabilities of these models to handle more diverse contexts, varying lengths of content, and a broader range of tasks. To achieve this, we meticulously curate a training dataset specifically designed for extremely long-form videos. Additionally, we further incorporate spatial-relevant knowledge through coarse correspondence markers among multi-frame visual inputs to make Oryx 3D-aware.

**Long-Form Temporal Training with Needle-In-A-Haystack.** The key ability for processing long-form video inputs is the identification of specific information within an extensive context, akin to the "needle-in-a-haystack" task in the NLP field. To enhance the Oryx model's capability to pinpoint details, we prepare long-form temporal needle-in-a-haystack training data. Specifically, we source video samples from the MovieNet (Huang et al., 2020) dataset, which comprises an average of 1000 frames per movie and an average duration of 45 minutes, thereby providing a natural setting for retrieving designated targets. We devise two tasks to train the model: captioning and differing. The captioning task requires the model to generate captions for frames at specific indices, while the differing task involves identifying differences between two frames given their indices. The training corpus is generated using SOTA LLMs, which produces captions for single frames or frame pairs. These captioned frames are then reinserted into the overall movie sequences, ensuring the training data maintains contextual integrity.

**Learning Spatial-Aware Knowledge via Coarse Correspondences.** Recent advancements have focused on enhancing multi-modal LLMs with 3D understanding capabilities. These approaches primarily treat 3D tasks as multi-image inputs. However, unlike video inputs, multi-view images generated from 3D environments lack temporal or trajectory cues, which are essential for MLLMs to

accurately process sequential data. As a result, previous methods often struggle to achieve correct spatial understanding when evaluated against 3D benchmarks.

Building on the work of (Liu et al., 2024a), we introduce coarse correspondences into our training dataset. The core concept is to assign a consistent label to the same object across different frames, allowing the model to better capture spatial correlations across multiple views. This approach aims to enhance the model's ability to develop a more accurate 3D spatial understanding. Specifically, we utilize Track-Anything (Yang et al., 2023b) as our tracking model to generate coarse correspondences for the ScanQA training set. These data are then incorporated into the final training set.

### 3.2.2 TRAINING PIPELINE & DATA MIXTURE

The training pipeline of Oryx is lightweight and direct in a 2-stage strategy. We start from a well-trained vision tower OryxViT and a Large Language Model. The first stage involves only image data following common practice (Liu et al., 2024d;b). The second stage uses a mixture of data from images, videos, and corresponding 3D frames and we train the multi-source data jointly thanks to our unified design. All of our training data are collected from open-source datasets, therefore ensuring the reproducibility of the Oryx model and holding room for improvement with better data curation.

**Stage 1: Text-Image Pre-training and Supervised Fine-tuning.** In the first stage of our training process, we focus on developing the foundational vision-language capabilities of the Oryx model using image data. This stage begins with a pre-training phase to train the dynamic compressor component with the basic image captioning data. Following this, we gather a collection of 4 million supervised fine-tuning image-text pairs that focus on high-quality knowledge learning. This data is sourced from various open-source academic datasets, including LLaVA-NeXt (Liu et al., 2024c), Cauldron (Laurençon et al., 2024), and Cambrian-1 (Tong et al., 2024). It is important to note that we do not incorporate large-scale pre-training stages as described in (Li et al., 2024a) or employ exclusive supervised fine-tuning data such as those in (Lin et al., 2023b; Bai et al., 2023), as our primary objective is to validate the effectiveness of our unified Oryx architecture.

**Stage 2: Joint Supervised Fine-tuning.** In Stage 2, we further conduct a supervised fine-tuning procedure following the initial stage, aiming to jointly train the Oryx model with image, video, and 3D-aware visual inputs. The image training data is sampled from the dataset collected during the supervised fine-tuning phase of Stage 1, ensuring a balanced ratio of image and video data. For video data, we source both comprehensive and multiple-choice datasets from open-source video repositories. Comprehensive datasets, which include question-answering and captioning tasks, are integrated using VideoChatGPT-Plus (Maaz et al., 2024), ShareGPT4Video (Chen et al., 2024a) and LLaVA-Hound (Zhang et al., 2024b). To enhance performance on multiple-choice benchmarks, we further incorporated Cinepile (Rawal et al., 2024), NextQA (Xiao et al., 2021) and PerceptionTest (Patraucean et al., 2024) into our training dataset. Additionally, we include video samples of needle-in-a-haystack data generated by GPT-4o (OpenAI, 2024) for long-form video learning and spatial-aware 3D multi-frame samples from the ScanQA (Azuma et al., 2022) training dataset, culminating in a total of around 650k video samples. The supervised fine-tuning strategy in this stage mirrored that of Stage 1, ensuring consistency in the training approach.

## 4 EXPERIMENTS

We conduct comprehensive experiments across multiple vision-language benchmarks to demonstrate the effectiveness of our method. In this section, we present the main results on general video understanding (Sec. 4.1), long-form video benchmarks (Sec. 4.2), 2D & 3D spatial understanding (Sec. 4.3). Finally, we provide analysis experiments and critical ablation studies on design elements.

### 4.1 GENERAL TEMPORAL UNDERSTANDING

**Setup.** We present the experimental results on general multi-modal video understanding datasets, as video data provides comprehensive insights into visual-language abilities, especially when dealing with complex and diverse visual inputs. We select several representative and popular benchmarks, encompassing both multiple-choice and generation tasks for evaluation. We conduct evaluations on four multiple-choice benchmarks. VideoMME (Fu et al., 2024) performs a full spectrum of diverse

Table 1: **General Temporal Understanding.** We conduct experiments on four multiple-choice benchmarks and three generation benchmarks comprehensively and report the main score for each dataset. Oryx exhibits superior performance under a wide range of open-sourced video MLLMs.

| Model | Size | VideoMME | NextQA | MVBench | PercepTest | MMB-Video | VCG | VDC |
|---|---|---|---|---|---|---|---|---|
| *Proprietary Models* | | | | | | | | |
| GPT-4V (OpenAI, 2023b) | - | 59.9/63.3 | - | 43.7 | - | 1.53 | 4.06 | 4.00 |
| GPT-4o (OpenAI, 2024) | - | 71.9/77.2 | - | - | - | 1.63 | - | - |
| Gemini-1.5-Pro (GeminiTeam, 2024) | - | 75.0/81.3 | - | - | - | 1.30 | - | - |
| *Open-Sourced Video MLLMs* | | | | | | | | |
| VideoChat2-HD (Li et al., 2024c) | 7B | 45.3/55.7 | 79.5 | 62.3 | 47.3 | 1.18 | 3.10 | - |
| VideoLLaMA2 (Cheng et al., 2024) | 7B | 47.9/50.3 | - | 54.6 | 51.4 | - | 3.13 | - |
| LLaVA-OneVision (Li et al., 2024a) | 7B | 58.2/61.5 | 79.4 | 56.7 | 49.7 | - | 3.51 | 3.75 |
| Kangaroo (Liu et al., 2024e) | 8B | 56.0/57.6 | - | 61.1 | - | 1.44 | - | - |
| VideoCCAM (Fei et al., 2024) | 9B | 53.9/56.1 | - | 64.6 | - | - | - | - |
| LLaVA-Next-Video (Zhang et al., 2024c) | 34B | 52.0/54.9 | 70.2 | - | 51.6 | - | 3.34 | 3.48 |
| PLLaVA (Xu et al., 2024a) | 34B | - | - | 58.1 | - | - | 3.48 | - |
| VILA-1.5 (Lin et al., 2023b) | 40B | 60.1/61.1 | 67.9 | - | 54.0 | - | 3.36 | 3.37 |
| VideoLLaMA2 (Cheng et al., 2024) | 72B | 61.4/63.1 | - | 62.0 | 57.5 | - | 3.16 | - |
| LLaVA-OneVision (Li et al., 2024a) | 72B | 66.2/69.5 | 80.2 | 59.4 | 66.9 | - | 3.62 | 3.60 |
| Oryx | 7B | 58.3/62.6 | 81.9 | 63.9 | 68.6 | 1.47 | 3.53 | **3.76** |
| Oryx | 34B | 63.2/67.4 | 83.5 | 64.7 | 71.4 | 1.49 | 3.51 | 3.66 |
| Oryx-1.5 | 7B | 58.8/64.2 | 81.8 | 67.6 | 70.0 | 1.49 | 3.62 | 3.74 |
| Oryx-1.5 | 32B | **67.3/74.9** | **85.0** | **70.1** | **74.0** | **1.52** | **3.66** | 3.63 |

videos and varying temporal lengths. NextQA (Xiao et al., 2021) is a classic benchmark for video reasoning. MVBench (Li et al., 2024c) performs 20 challenging video tasks for video comprehension. Perception Test (Patraucean et al., 2024) focuses on the perception and reasoning skills of MLLMs. For generation-relevant benchmarks scored by advanced proprietary models, we integrate evaluations on MMBench-Video (Fang et al., 2024), Video-ChatGPT(VCG) (Maaz et al., 2023), and Video Detailed Caption(VDC) benchmarks. Following common practice, GPT-4-1106 (OpenAI, 2023c) is used as the evaluator for MMBench-Video (Fang et al., 2024), GPT-3.5-0613 (OpenAI, 2023a) is employed for Video-ChatGPT (Maaz et al., 2023) and Video Detailed Caption.

**Results.** The experimental results, as detailed in Table 1, demonstrate that the Oryx model achieves highly competitive outcomes in general video understanding tasks. We surpass a broad spectrum of near-term video-specific MLLMs and establish new state-of-the-art. The Oryx model attains tier-1 performance among small-sized MLLMs (approximately 7B parameters) and exhibits competitive performance when compared to larger MLLMs (exceeding 30B parameters), even rivaling models with 72B parameters. On the VideoMME benchmark (Fu et al., 2024) with subtitles, the Oryx-1.5 models achieve mean accuracies of 64.2 and 74.9 for 7B and 32B variants. Oryx-1.5 also demonstrates robust performance across various multiple-choice datasets by surpassing previous state-of-the-art results by 4.8% and 7.1% on NextQA (Xiao et al., 2021) and Perception Test (Patraucean et al., 2024). Additionally, the Oryx model performs convincingly on GPT-eval benchmarks, with an average score of 1.52 on MMBench-Video (Fang et al., 2024), 3.66 and 3.76 on VideoChatGPT (Maaz et al., 2023) and Video Detailed Caption, respectively. Remarkably, the Oryx model outperforms advanced proprietary models such as GPT-4V (OpenAI, 2023b) and Gemini-1.5-Pro (GeminiTeam, 2024) on several of the most challenging benchmarks.

## 4.2 LONG-FORM TEMPORAL UNDERSTANDING

**Setup.** To further demonstrate the long-context understanding capability, we conduct experiments on benchmarks specifically designed for long video evaluation. We select three mainstream benchmarks specifically designed for long video understanding for a comprehensive evaluation. MLVU (Zhou et al., 2024), encompasses videos ranging from 3 minutes to 2 hours and includes 9 distinct tasks that assess both global and local information within the video content. LongVideoBench (Wu et al., 2024) presents a primary challenge of retrieving and reasoning over a dataset comprising 3k long video inputs. Additionally, we utilize the long video subset of the VideoMME (Fu et al., 2024) benchmark, which features videos with lengths ranging from 30 minutes to 60 minutes.

Table 2: **Long-Form Temporal Understanding.** We show results on three mainstream long-form temporal understanding datasets, each featuring video inputs of tens of minutes in duration. Oryx demonstrates superior performance, achieving state-of-the-art results and surpassing several proprietary models across various benchmarks.

| Model | Size | MLVU | LongVideoBench | VideoMME-Long | |
| --- | --- | --- | --- | --- | --- |
| | | | | w/o subs | w subs |
| *Proprietary Models* | | | | | |
| GPT-4V (OpenAI, 2023b) | - | 49.2 | 60.7 | 53.5 | 56.9 |
| GPT-4o (OpenAI, 2024) | - | 64.6 | 66.7 | 65.3 | 72.1 |
| Gemini-1.5-Pro (GeminiTeam, 2024) | - | - | 64.4 | 67.4 | 77.4 |
| *Open-Sourced Video MLLMs* | | | | | |
| VideoLLaMA2 (Cheng et al., 2024) | 7B | 48.5 | - | 42.1 | 43.8 |
| LongVA (Zhang et al., 2024a) | 7B | 56.3 | - | 46.2 | 47.6 |
| LLaVA-OneVision (Li et al., 2024a) | 7B | 64.7 | - | - | - |
| Kangaroo (Liu et al., 2024e) | 8B | 61.0 | 54.8 | 46.6 | 49.3 |
| LongVILA (Xue et al., 2024) | 8B | - | - | 39.7 | - |
| VideoCCAM (Fei et al., 2024) | 14B | 63.1 | - | 46.7 | 49.9 |
| LLaVA-Next-Video (Zhang et al., 2024c) | 34B | - | 50.5 | - | - |
| PLLaVA (Xu et al., 2024a) | 34B | - | 53.2 | - | - |
| VILA-1.5 (Lin et al., 2023b) | 40B | 56.7 | - | 53.8 | 55.7 |
| LLaVA-OneVision (Li et al., 2024a) | 72B | 66.4 | 61.3 | **60.0** | 62.4 |
| Oryx | 7B | 67.5 | 55.3 | 50.3 | 55.8 |
| Oryx | 34B | 70.8 | **62.2** | 53.9 | 58.0 |
| Oryx-1.5 | 7B | 67.5 | 56.3 | 51.2 | 58.3 |
| Oryx-1.5 | 32B | **72.3** | 62.0 | 59.1 | **69.7** |

Table 3: **Image Understanding.** We conduct 2D spatial understanding tasks on six representative image benchmarks, including general and task-specific benchmarks. Our Oryx model achieves tier-1 performance across a wide range of MLLMs.

| Model | Size | MMBench | MMMU | DocVQA | OCRBench | AI2D | TextVQA |
| --- | --- | --- | --- | --- | --- | --- | --- |
| Deepseek-VL (Lu et al., 2024) | 7B | 73.2 | 36.6 | - | 456 | - | 64.7 |
| Monkey (Li et al., 2024d) | 7B | 72.4 | 40.7 | - | 534 | 68.5 | - |
| LLaVA-NeXT (Liu et al., 2024c) | 8B | 72.1 | 41.7 | 78.2 | 531 | 71.6 | - |
| Bunny-LLama3 (He et al., 2024) | 8B | 77.2 | 43.3 | - | 444 | 69.4 | - |
| Cambrian-1 (Tong et al., 2024) | 8B | 75.9 | 42.7 | 77.8 | 624 | 73.6 | 71.7 |
| VILA-1.5 (Lin et al., 2023b) | 8B | 75.3 | 38.6 | - | - | - | 68.5 |
| Idefics2 (Laurençon et al., 2024) | 8B | 76.7 | 43.0 | - | - | - | 73.0 |
| Yi-VL (Young et al., 2024) | 34B | - | 45.1 | - | 290 | 65.9 | - |
| LLaVA-NeXT (Liu et al., 2024c) | 34B | 79.3 | 49.7 | 84.0 | 574 | 74.9 | - |
| Cambrian-1 (Tong et al., 2024) | 34B | 81.4 | 49.7 | 75.5 | 600 | 79.7 | 76.7 |
| VILA-1.5 (Lin et al., 2023b) | 40B | 82.4 | 51.9 | - | - | - | 73.4 |
| LLaVA-OneVision (Li et al., 2024a) | 72B | 85.6 | 56.8 | 93.1 | - | 85.6 | - |
| Oryx | 7B | 81.4 | 43.9 | 89.0 | 672 | 78.5 | 75.0 |
| Oryx | 34B | 84.5 | 50.3 | 91.4 | 743 | 81.0 | 77.8 |
| Oryx-1.5 | 7B | 81.3 | 47.1 | 90.1 | 713 | 79.7 | 75.7 |
| Oryx-1.5 | 32B | **86.3** | **56.1** | **92.7** | **746** | **83.2** | **78.3** |

**Results.** Results are shown in Table 2, which highlights the efficacy of our unified and on-demand design across varying temporal lengths and our further efforts in the context of long video retrieval. The Oryx model exhibits a remarkable capability in understanding long-form video content. Specifically, our Oryx-1.5-7B model surpasses all existing 7B model series on long video benchmarks. Furthermore, the Oryx-1.5-32B model showcases strong performance across larger MLLMs, achieving a mean accuracy improvement of 5.9% and 0.7% over previous state-of-the-art models equipped with 72B parameter LLMs on the MLVU (Zhou et al., 2024) and LongVideoBench (Wu et al., 2024) benchmarks, respectively. Notably, the Oryx-1.5-32B model also outperforms GPT-4o (OpenAI, 2024) on the challenging MLVU (Zhou et al., 2024) benchmark by a margin of 7.7%, underscoring its advanced capabilities in long video understanding.

### 4.3 2D & 3D SPATIAL UNDERSTANDING

As we perform a general solution across spatial and temporal understanding, we incorporate both image and 3D benchmarks in our assessments to show the foundation multi-modal capabilities of Oryx model and the potential for extending to more visual tasks, formats, and circumstances.

Table 4: **3D Spatial Understanding.** We use the popular ScanQA (Azuma et al., 2022) dataset and evaluate the relevant scores. We compare the Oryx model with 3D-specific models together with general open-source MLLMs. Oryx excels in 3D spatial understanding tasks, highlighting its versatility across various applications.

| Model | Size | METEOR | ROUGE-L | CIDEr | BLEU-1 | BLEU-2 |
|---|---|---|---|---|---|---|
| *3D-Specific Models* | | | | | | |
| VoteNet+MCAN (Qi et al., 2019) | - | 11.4 | 29.8 | 54.7 | 28.0 | 16.7 |
| ScanQA (Azuma et al., 2022) | - | 11.5 | 30 | 55.4 | 26.9 | 16.6 |
| ScanRefer+MCAN (Chen et al., 2020) | - | 13.1 | 33.3 | 64.9 | 30.2 | 20.4 |
| 3D-LLM (Hong et al., 2023) | - | 14.5 | 35.7 | 69.4 | 39.3 | 25.2 |
| *General Open-Source MLLMs* | | | | | | |
| BLIP2 (Li et al., 2023) | - | 11.3 | 26.6 | 45.7 | 29.7 | 16.2 |
| Flamingo (Alayrac et al., 2022) | 7B | 11.3 | 31.1 | 55 | 25.6 | 15.2 |
| Mantis (Jiang et al., 2024) | 7B | - | 16.1 | - | - | - |
| LLaVA-Next-Interleave (Li et al., 2024b) | 14B | - | 34.5 | - | - | - |
| LLaVA-OneVision (Li et al., 2024a) | 72B | - | 35.8 | - | - | - |
| Oryx | 7B | 14.5 | 35.5 | 69.1 | 35.8 | 24.4 |
| Oryx | 34B | 15.0 | 37.3 | 72.3 | **39.6** | **26.7** |
| Oryx-1.5 | 7B | 15.2 | 38.4 | 73.5 | 38.6 | 24.6 |
| Oryx-1.5 | 32B | **15.3** | **38.4** | **74.3** | 38.8 | 24.4 |

**Image Benchmarks.** We select a diverse set of mainstream and representative image benchmarks to evaluate the model's proficiency in image understanding. Specifically, we included MMBench (Liu et al., 2023a) and MMMU (Yue et al., 2024) to assess general image understanding capabilities, and DocVQA (Mathew et al., 2021), OCRBench (Liu et al., 2023b), AI2D (Kembhavi et al., 2016), and TextVQA (Singh et al., 2019) to evaluate the model's performance on specific tasks such as document recognition, OCR, text understanding tasks, etc. The results are summarized in Table 3. Notably, the Oryx model maintains pioneering results on image benchmarks, such as an 86.3% mean accuracy on MMBench (Liu et al., 2023a) and a 92.7% accuracy on DocVQA (Mathew et al., 2021). Such results demonstrate the effectiveness of our method in comprehending images with more simple and lightweight training pipelines, data curation, and strategies compared with concurrent works.

**3D Spatial Understanding.** We conduct 3D spatial understanding on the classic ScanQA validation set, following the protocol established by previous work (Azuma et al., 2022; Hong et al., 2023; Liu et al., 2024a). We incorporate advanced baseline models, including 3D-specific models and general open-source MLLMs supporting 3D spatial tasks for a comprehensive comparison. As shown in Table 4, the Oryx model not only outperforms previous specialized models designed for 3D understanding, but also surpasses the recently updated general MLLMs and specially designed 3D-LLM (Hong et al., 2023). These results underscore the robust adaptability of our method in addressing 3D spatial tasks.

## 4.4 ANALYSIS

**Effects of resolution and resize strategy across benchmarks.** To illustrate the effectiveness of the advanced native representation for visual inputs, we conduct ablation analysis on the effects of resolution in Figure 3. We compare inputs with native resolution to inputs rescaled to specific overall number of pixels while maintaining the original aspect ratios. The left figure presents the scores on several benchmarks, where we utilize images scaled to $768^2$ pixels, $1024^2$ pixels, images with native resolution, and larger images (2x area) with native resolution. The results indicate that native resolution consistently outperforms fixed sizes, with the performance gap becoming more pronounced in the DocVQA and OCRBench datasets. These datasets require the visual encoder to process more natural image inputs for text understanding. Additionally, further enlarging the resolution does not yield significant gains in most benchmarks. The right figure illustrates the performance trends on MMBench (Liu et al., 2023a) and OCRBench (Liu et al., 2023b) with varying visual input resolutions. Our findings suggest that while larger images generally lead to better performance, maintaining the native resolution emerges as a simple yet effective strategy for optimizing performance.

**Effectiveness of the Oryx Architecture.** We conduct more ablation experiments on the design of the Oryx architecture in Table 5. For the visual representation, we compare OryxViT with the mainstream SigLIP visual encoder. Our comparison highlights the superior alignment performance of OryxViT. Additionally, we fairly compare previous dynamic partition approaches with visual

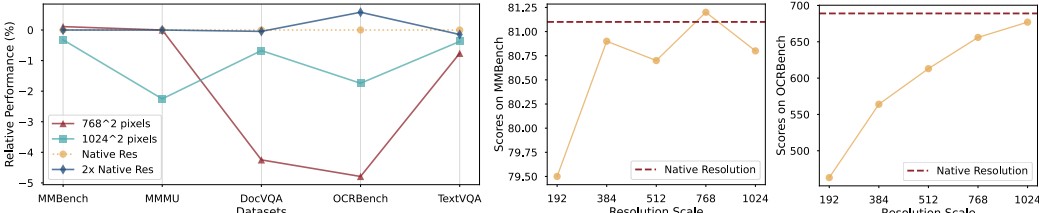

Figure 3: **Effects of resolution and resize strategy across benchmarks.** The left figure shows the performance across benchmarks with fixed size, native size, and larger images. The right figure shows the trend of performance with varying resolutions, where we illustrate the performance of native resolution for reference. The text-relative benchmarks show more sensitivity to the resolution scale, while all the benchmarks benefit from the visual inputs with native resolution.

Table 5: **Ablations on the Oryx Architecture.** We evaluate our design of two core architectures within the Oryx model. (a) examines the impact of the visual encoder and the method of processing visual inputs, demonstrating the superiority of native visual representations compared with dynamic partition and the strong visual-text alignment capability of OryxViT. (b) assesses the influence of dynamic compression modules in comparison to conventional MLP connectors, revealing significant performance gains due to improved fusion of image and video data. Various downsampling approaches were tested, with average pooling yielding the best performance.

(a) Ablation study on Visual Encoder.

| Visual Enc. | Res. | DocVQA | OCRBench | MMBench |
|---|---|---|---|---|
| SigLIP | Partition | 74.8 | 531 | 68.0 |
| SigLIP | Native | 17.1 | 67 | 15.8 |
| OryxViT | Partition | 76.3 | 549 | 68.9 |
| OryxViT | Native | 78.5 | 572 | 69.3 |
| OryxViT | Optimal | 79.2 | 572 | 69.9 |

(b) Ablation Study on Compression Module.

| Connector | Downsample | VideoMME | MLVU |
|---|---|---|---|
| MLP | Avg Pool | 54.6 | 57.5 |
| Dy.Compressor | Avg Pool | 55.4 | 59.3 |
| Dy.Compressor | DWConv | 55.0 | 58.9 |
| Dy.Compressor | Conv-MLP | 54.7 | 58.5 |

inputs of native resolutions. We conclude from the results that the previous mainstream multi-modal encoder SigLIP (Zhai et al., 2023) fails to process native visual input and only works on fixed resolution with the dynamic partition trick. On the contrary, the OryxViT benefits from the visual inputs at native resolution, which is superior to the partition approach. As an arbitrary visual encoder, we are also curious about the limit of resolutions, where we find that searching for the optimal anchor resolution leads to better performance (the last line in Table 5 (a)). However, for the sake of fairness and efficiency, we do not employ this optimization in our primary evaluations. We report our results on several representative image benchmarks, including DocVQA, OCRBench, and the general MMBench datasets, using a subset of image training data for efficient training.

For the connector module, we compare the proposed dynamic compressor with the straightforward MLP architecture. The dynamic compressor demonstrates superior performance on both general and long temporal benchmarks by better fusing multi-modal data. Furthermore, our analysis reveals that average pooling yields better results for higher compression visual inputs compared to parameter-reliant approaches such as DWConv and Conv-MLP. This improvement is likely due to the parameter-free nature of average pooling, which preserves the distribution of visual features, and more complex downsampling layers may not be effectively trained through the current pipeline. Our analysis of the connector module is conducted on a subset of video training data to maintain training efficiency.

## 5 CONCLUSION

In this paper, we introduce the Oryx series, a novel approach designed to handle diverse visual inputs across varying tasks, temporal lengths, and resolutions in an on-demand manner. The Oryx model stands out as a unified multi-modal framework for spatial-temporal understanding, leveraging the innovative design of OryxViT for native resolution processing, the Dynamic Compressor for efficient data compression, and a robust joint training strategy. Our extensive evaluations demonstrate that the Oryx model achieves outstanding performance across a wide array of image, video, and 3D mainstream benchmarks. We hope that our work offers a novel perspective on multi-modal learning and paves the way for the development of more general MLLMs in future research endeavors.

## ACKNOWLEDGEMENT

This work was supported in part by the National Natural Science Foundation of China under Grant 62125603, Grant 62336004, Grant 62321005, and Grant 62441616, and in part by the Beijing Natural Science Foundation under Grant No. L247009.

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

APPENDIX

## A  GENERATION RESULTS

**Video Summarization and Detailed Description.** As shown in Fig. 4, the Oryx model effectively generates a comprehensive and detailed caption that accurately summarizes the input video. It captures the main event while preserving essential information. Oryx produces more accurate results about the match information, the name, and the status of the player. In contrast, LLaVA-OneVision (Li et al., 2024a) shows the wrong name, and LongVILA tells the wrong score on the board.

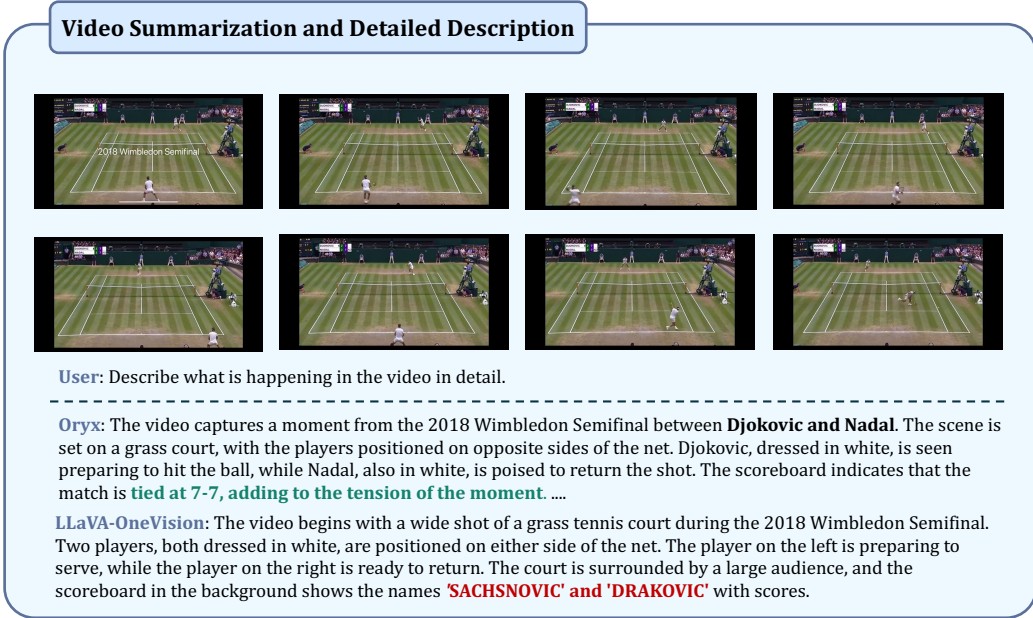

Figure 4: Oryx is able to make a comprehensive video summary and detailed caption.

**Video Multiple Choice and Reasoning.** Oryx is also capable of reasoning based on the input video. As demonstrated in Fig. 5, Oryx can answer questions through analogy and generate well-reasoned responses.

**Skill Learning From Videos.** Oryx can acquire useful skills from the input video. As demonstrated in Fig. 6, Oryx learns to use Google Scholar to cite a paper by following the steps shown in the video. It illustrates all the necessary steps to complete the citation, highlighting its strong skill-learning capability and potential for agent-based tasks and task execution. Although the baseline model provides additional instructions, some detailed steps are not depicted in the original video. We believe this hallucination may stem from the information in the training data.

**Understanding 3D with Coarse Correspondences.** Oryx enhances its 3D spatial understanding using coarse correspondences. Fig. 7 illustrates Oryx's reasoning process, demonstrating its ability to improve 3D comprehension through these correspondences and generate accurate reasoning outcomes. In the challenging task involving direction in a first-person view, LLaVA-OneVision (Li et al., 2024a) provides incorrect conclusions.

## B  FAILURE CASES

In this section, we further test the Oryx model on more challenging samples. We provide some representative failure cases to show the limitations of the Oryx model and point out the future direction for VLMs. The incorrect response is highlighted in red.

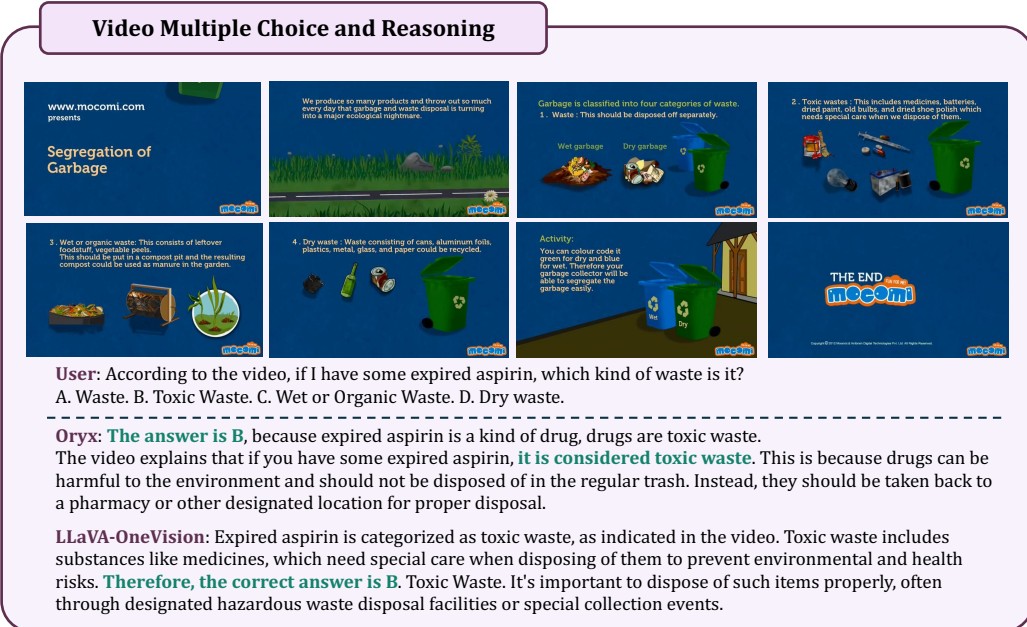

Figure 5: Oryx learns to reason through the input video.

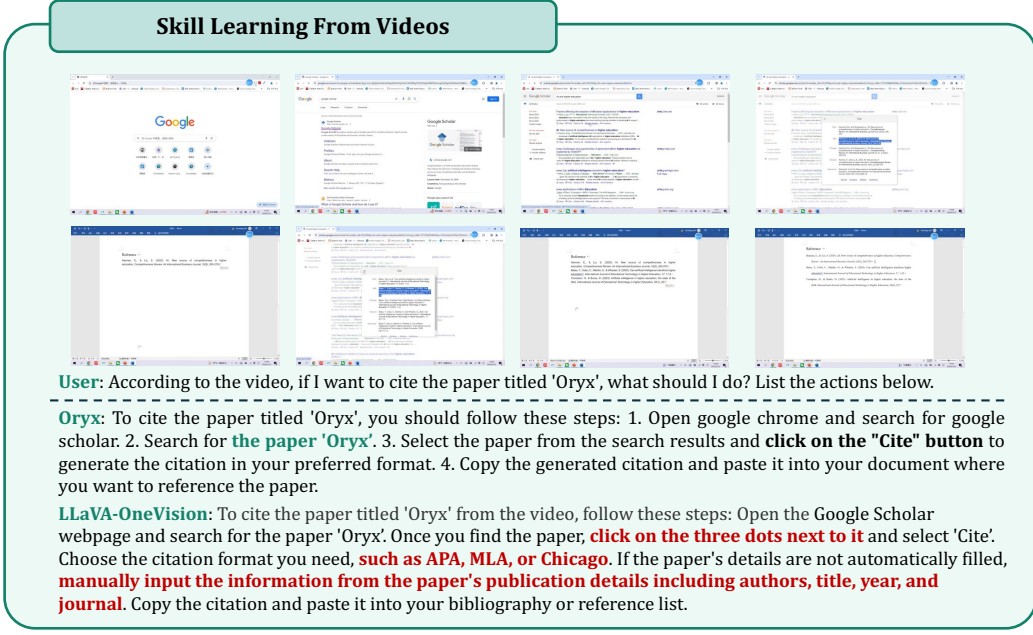

Figure 6: Oryx learns useful skills from the input video.

**Time Perception.** We provide an example of time perception in Fig. 8. From the responses, we can observe that current video understanding models using uniform frame sampling tend to lose the temporal context in videos, which is critical information between frames. For instance, Oryx provides incorrect answers and cannot accurately determine the timestamps for the given video. We believe that incorporating timestamps directly into the video or processing video inputs without frame sampling may help address this issue, leading to a more comprehensive video understanding capability.

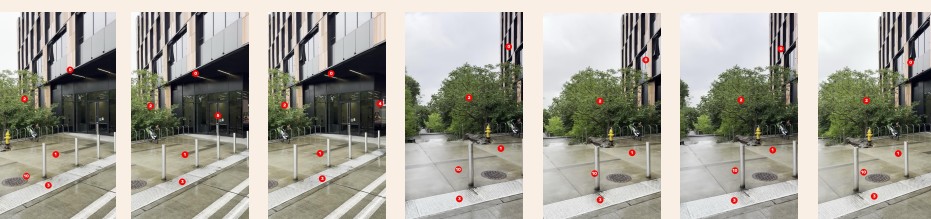

**Understanding 3D with Coarse Correspondences**

**User**: If Alice comes out of the building and wants to go down the stairs and cross the street using the pedestrian crossing, which direction should she turn after exiting the door?
A. Left B. Right.
Please answer from Alice's perspective, not the observer's. Explain why using the labels inside the images.

-----------------------------------------------------

**Oryx**: If Alice comes out of the building and wants to go down the stairs and cross the street using the pedestrian crossing, she should **turn right after exiting the door**. This is because the pedestrian crossing is located on the right side of the entrance when facing it from inside the building. The labels inside the images indicate that the pedestrian crossing (label 3) is to the right of the door (label 4), and the stairs (label 1) are also situated to the right of the door when viewed from Alice's perspective. Therefore, turning right would lead Alice towards the pedestrian crossing and then across the street.

**LLaVA-OneVision**: After exiting the door, Alice should **turn left**. This is because… , as indicated by the position of the yellow fire hydrant (label 1) and the bicycle rack (label 2) in the images. Turning left will lead her towards these features and ultimately to the street where she can use the pedestrian crossing.

Figure 7: Oryx understands 3D spatial information through coarse correspondences.

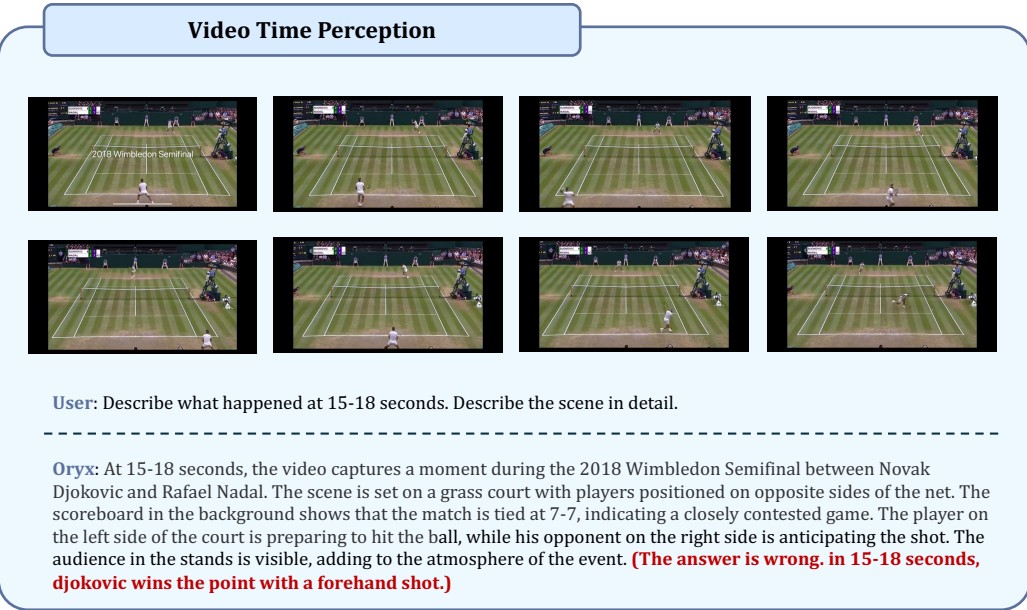

**Video Time Perception**

**User**: Describe what happened at 15-18 seconds. Describe the scene in detail.

-----------------------------------------------------

**Oryx**: At 15-18 seconds, the video captures a moment during the 2018 Wimbledon Semifinal between Novak Djokovic and Rafael Nadal. The scene is set on a grass court with players positioned on opposite sides of the net. The scoreboard in the background shows that the match is tied at 7-7, indicating a closely contested game. The player on the left side of the court is preparing to hit the ball, while his opponent on the right side is anticipating the shot. The audience in the stands is visible, adding to the atmosphere of the event. **(The answer is wrong. in 15-18 seconds, djokovic wins the point with a forehand shot.)**

Figure 8: Failure cases about time perception.

**Knowledge Reasoning and Transfer.** We provide the knowledge reasoning video in Fig. 9. From the responses, we can observe that the Oryx model lacks knowledge of chemical reactions, indicating that the training data of current MLLMs is not comprehensive enough. We believe that expanding the knowledge base of current MLLMs is an urgent issue. Additionally, this example requires simple reasoning to integrate chemical knowledge into the main idea of the video, which poses a challenge for Oryx. Enhancing the reasoning capability may help address this issue.

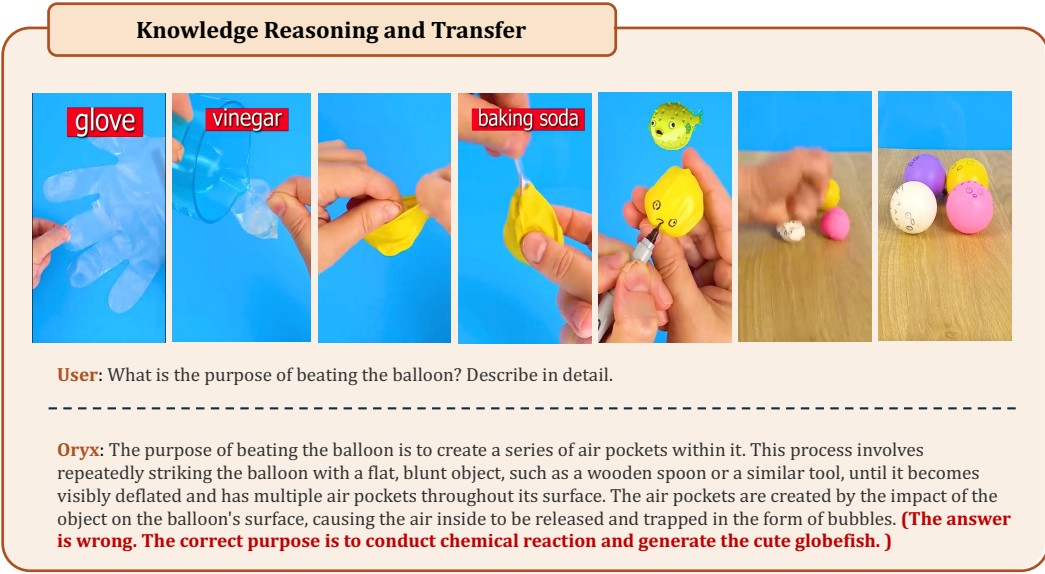

Figure 9: Failure cases about knowledge reasoning and transfer.

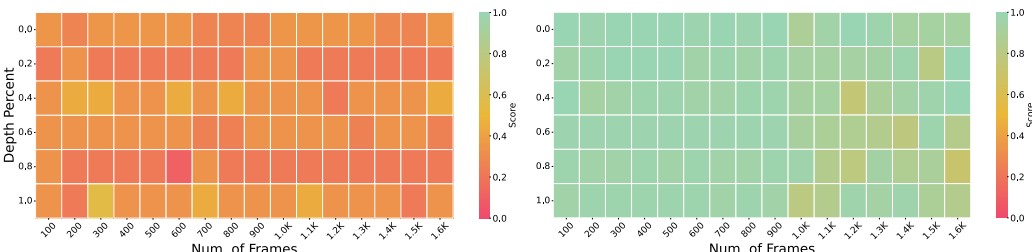

Figure 10: **Visualization Results on Video Needle-In-A-Haystack Experiments.** We compare Oryx-7B (right subfigure) with LLaVA-Next-Video-7B (left subfigure) on the frame retrieval task. The results are shown for inserted depths ranging from 0.0 to 1.0 and the number of frames ranging from 0.1k to 1.6k. The Oryx model demonstrates superior performance in long-form understanding tasks, providing precise results even when a single relevant frame is embedded within over 1k frames of irrelevant information.

## C  MORE ANALYSIS

### C.1  VIDEO NEEDLE-IN-A-HAYSTACK

To demonstrate the retrieval ability in long-form visual inputs and test the quality of the dynamic compression module, we design the video needle-in-a-haystack experiment under extreme conditions, following the methodologies established in previous work (Zhang et al., 2024a; Xue et al., 2024). For this experiment, we select an extremely long video and then insert irrelevant image question-answering data as a single frame at arbitrary depths within the video. The model is tasked with answering questions related to these inserted images. We utilize LLaVA-Next-Video (Zhang et al., 2024c) of comparable size as our baseline. As depicted in Figure 10, baseline models trained with 32 frames failed to identify the images, suffering from severe information loss. In contrast, our method successfully retrieves the inserted images and accurately answers the questions, even with frame counts of 1.6k. This outcome strongly demonstrates the model's ability in long-form temporal understanding, facilitated by the on-demand compression module.

Table 6: **Test on Inference Speed and Memory Cost.**

| Backbone | Processing Approach | Throughput (image/s) | Max Memory Cost |
|---|---|---|---|
| OryxViT | Native Resolution | 146.5 | 49.1GB |
| SigLIP | Dynamic Partition | 157.7 | 48.7GB |

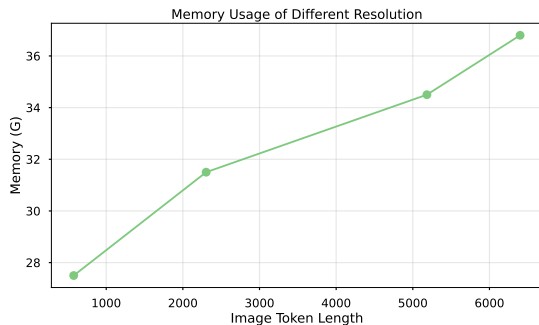

Figure 11: **Memory-Resolution Curve for OryxViT.**

## C.2    INFERENCE SPEED AND EFFICIENCY

We implement variable-length self-attention using the highly optimized FlashAttention (Dao et al., 2022) library. This allows the inference throughput of our arbitrary-resolution visual encoder to remain comparable to the dynamic partition approach used in previous methods. Additionally, the memory overhead and inference throughput remain negligible, as the variable-length attention operation is fully optimized through modifications in the CUDA kernels.

Moreover, the model size of the Vision Transformer is considerably smaller than that of large language models (400M parameters compared to 7B/32B). Consequently, the main memory cost arises from the weights and features of the LLMs, as full attention is computed on visual tokens within the LLM, even when using dynamic partitioning in visual encoding. Therefore, we maintain a similar efficiency to previous solutions in terms of inference speed and memory cost.

We tested the inference speed with an input image size of $1280 \times 1280$ on one NVIDIA A100 GPU. We observed that OryxViT is only 7% slower than SigLIP with the dynamic partition approach. We believe this overhead is acceptable given the improved performance and the ability to process images at their native resolutions directly. For the max memory cost, we set the batch size to 4 and the image size to a total of $1280 \times 1280$ pixels, with the aspect ratio randomly determined. The dynamic-partition baseline uses an average of 48.7G of memory, while OryxViT uses 49.1G, showing comparable results.

Additionally, we plotted the memory-resolution curve to illustrate how the memory footprint increases with varying resolutions in Fig. 11. Our experiment was conducted on an NVIDIA A100 GPU, using square-shaped input images. The results show that the memory cost is positively correlated with image resolution and is approximately in linear relation among common image resolutions, indicating that OryxViT provides a memory-efficient solution when scaling input resolutions.

Furthermore, we can also observe that the primary memory cost arises from the model weights and the features themselves. Notably, considering that the main memory cost comes from the computation in LLMs, our method with native resolution support will not introduce much extra memory footprint compared to previous methods like LLaVA-Next (Liu et al., 2024c) and InternVL2 (Chen et al., 2024b). Oryx can largely save memory if we apply dynamic compression on visual tokens (e.g., 2x or 4x downsampling) while previous methods do not support this feature.

## C.3 ABLATIONS ON TRAINING DATA

We conduct ablation experiments on the collected training data including the long-form video data and the 3D coarse corresponding data in Tab. 7. We can observe from the results that the long-form temporal data benefits the long video benchmarks. For 3D-related tasks, our proposed approach based on coarse correspondence is an effective solution for 3D benchmarks. We conducted experiments and provided results on ScanQA. "3D Data" indicates whether 3D datasets are included in the training set, while "C.C." refers to using the coarse correspondence approach to annotate objects in 3D-relevant videos. We observe that both components contribute to an improved understanding results.

Table 7: **Ablations on Training Data.**

(a) Effects on Long-Form Data.

| Long-Form Data | VideoMME | MLVU |
|:---:|:---:|:---:|
| ✗ | 55.2 | 58.1 |
| ✓ | **55.4** | **59.3** |

(b) Effects on 3D-relevant Data.

| 3D Data | C.C. | METEOR | ROUHE-L |
|:---:|:---:|:---:|:---:|
| ✗ | ✗ | 11.7 | 28.1 |
| ✓ | ✗ | 12.8 | 32.7 |
| ✓ | ✓ | **14.0** | **35.1** |

## C.4 DESIGNS FOR MLP ADAPTER

We explored using specialized projection layers when integrating video modality into our model. However, we found that employing a shared MLP for all visual inputs yields better performance. To align with this design, we use the Dynamic Compressor module to closely maintain the distribution of image and video features and employ the shared MLP for visual information fusion. This ensures that input tokens for LLMs maintain a consistent distribution for both images and videos, allowing the shared MLP to be better trained with joint visual knowledge.

We conducted experiments in Tab. 8 to support our hypothesis, comparing our shared MLP strategy with separate MLPs for images and videos. Both the image and video MLPs were initialized from pre-trained image weights. The results indicate that using separate MLPs negatively impacts video benchmarks, as the dual-projector design can lead to differing distributions for similar data. Therefore, using a single MLP is a more effective solution for visual encoding.

## C.5 ANALYSIS ON DOWNSAMPLING

**Analysis on Overall Downsampling Architecture.** We integrate several mainstream approaches, including direct average pooling, $2 \times 2$ spatial convolution, Q-former, and our proposed Dynamic Compressor for comparison. The results are presented in Tab. 9 (a). We reference results from VideoMME (Fu et al., 2024) and MLVU (Zhou et al., 2024). Our observations indicate that the proposed Dynamic Compressor outperforms traditional downsampling strategies based on average pooling and spatial convolution. Additionally, we find that Q-former-based methods are not suitable for handling long visual content with fixed lengths of visual tokens. This limitation arises because the information capacity of a visual token is directly proportional to its length, making fixed lengths inadequate for more complex cases involving longer visual content.

**Analysis on Downsampling Function in Dynamic Compressor** We are also curious which downsampling function performs better within the dynamic compressor. We compared average pooling, DWConv, and Conv-MLP architectures. The results are presented in Tab. 9 (b). Results indicate that the parameter-free average pooling outperforms parameter-dependent methods like DWConv and Conv-MLP. This improvement is likely because average pooling better preserves the distribution of visual features, whereas more complex downsampling layers may not be effectively trained with the current pipeline.

Table 8: **Ablations on Designs of MLP Adapter.**

| MLP Architecture | VideoMME | MLVU | MMBench | MMMU |
|---|---|---|---|---|
| Shared | 55.4 | 59.3 | 81.4 | 43.9 |
| Separated | 54.0 | 54.2 | 81.2 | 43.1 |

Table 9: **Analysis on Downsampling.**

(a) Overall Downsampling Architecture.

| Strategy | VideoMME | MLVU |
|---|---|---|
| Average Pooling | 54.6 | 57.5 |
| Convolution | 54.2 | 56.8 |
| Q-former | 42.7 | 35.3 |
| Dynamic Compressor | **55.4** | **59.3** |

(b) Downsampling Function in Dynamic Compressor.

| Function | VideoMME | MLVU |
|---|---|---|
| DWConv | 55.0 | 58.9 |
| Conv-MLP | 54.7 | 58.5 |
| Average Pooling | **55.4** | **59.3** |

## D    MORE DETAILS

### D.1    IMPLEMENTATION DETAILS

Our implementation integrates the Oryx model with multiple sets of LLMs, we use Qwen-2-7B (Qwen-Team, 2024a) and Yi-1.5-34B (Young et al., 2024) for Oryx, Qwen-2.5-7B (QwenTeam, 2024a) and Qwen-2.5-32B (QwenTeam, 2024a) for Oryx-1.5. For the visual encoder, we use our pre-trained OryxViT to support arbitrary-resolution visual inputs. During the pre-training stage, we utilize 558k captioning data from LLaVA-1.5 (Liu et al., 2024b), unfreezing the parameters of the dynamic compression module. The image SFT stage involves curating an open-source dataset of around 4M images. In the joint training stage, we incorporate approximately 1.2M data consisting of images sampled from the previous stage and video/3D data implemented in Sec. 3.2. For video data, we restrict the frame number to 64 for standard videos of low compression ratio and 256 for long videos of high compression ratio. We use the $2 \times 2$ average downsample for low compression and $4 \times 4$ average downsample for high compression. Image data are maintained at their native resolution, with a maximum size of 1536 pixels, while video data resolutions are confined to a range of 288 to 480 pixels. The rest of the training details are provided in the appendix.

### D.2    DETAILS OF ORYXVIT

We pre-train OryxViT with a relatively small language model (Qwen2-0.5B (QwenTeam, 2024a) in our implementation) to enhance the language interface and improve vision-language alignment. We unfreeze OryxViT and apply LoRA fine-tuning to the language models. As a result, the total number of trainable parameters is 0.6B, making the training process significantly faster than supervised fine-tuning in the main stage (approximately 10 times faster). We collected a total of 400M pre-training data, focusing mainly on image captioning and image OCR tasks. For image captioning, we used the CapsFusion (Yu et al., 2024) datasets, and for OCR tasks, we employed synthesized OCR data pairs with OCR models. We set the batch size to 2048 and used a similar cross-entropy loss as in the main stages.

### D.3    TRAINING DETAILS

**Stage 1.** For stage 1, we first pre-train the connector module between the visual encoder and Large Language Model for the initial alignment between image and text modalities. We conduct our experiments on 558k caption data from BLIP (Li et al., 2023) model following LLaVA-1.5 (Liu et al., 2024b). We only unfreeze the parameter for the connector while maintaining other parameters fixed. We adopt the total training batch size at 256 and the overall learning rate at 1e-3. We maintain the aspect ratio for the input image while adjusting the overall pixels to $768^2$ to reduce the computational cost. The training cost for the pre-training alignment is lightweight thanks to the small number of parameters for the connector and the relatively lower image-text data pairs. Subsequently, we conduct the supervised fine-tuning stage with 4.1M image data. We freeze the parameter for the visual encoder

while unfreezing the connector and the Large Language Model following common practice. In this stage, we use the native resolution of the image while restricting the maximum number of pixels at $1280^2$ for efficiency. For the image larger than $1280^2$ pixels, we scale down the image to match the overall number of pixels. We set the learning rate at 2e-5 for Oryx-7B and the learning rate at 1e-5 for Oryx-34B. We adopt the total batch size at 128 and conduct our experiments on 64 NVIDIA A100-40G GPUs for Oryx-7B and 64 NVIDIA A800-80G GPUs for Oryx-34B, as larger models need more GPU memories. The total model maximum length is set as 8192.

**Stage 2.** For stage 2, we continuously train the Oryx model from the multi-modal LLMs in stage 1. We randomly sample around 600k image data from the supervised fine-tuning stage in stage 1 and add an additional 650k temporal and 3D data from open-source multi-modal datasets, resulting in an overall number of 1.2M further supervised fine-tuning data. In the more general stage, we increase the restriction for image pixels to $1536^2$ to meet the longer sequential length in temporal data. We maintain the aspect ratio of video data while normalizing each frame to the minimum size of $288^2$ pixels and the maximum size of $480^2$ pixels, therefore the token length before the compression module ranges from 324 to 900. We adopt $1 \times 1$ path for the image data, $2 \times 2$ pooling path for the multi-frame data including video and 3D-relevant data, and $4 \times 4$ pooling path for the extremely long video needle-in-the-haystack retrieval data. We maintain most of the training hyper-parameters identical to stage 1, with a total batch size of 128, a learning rate of 2e-5 for Oryx-7B, and a learning rate of 1e-5 for Oryx-34B and Oryx-1.5-32B. We sample 1 frame per second for video data and set the max frame number to 64 frames. We uniformly sample the frames among all the frames if the number exceeds the upper bound. The maximum sequence length is set to 16384.

