# OpenReview forum: "Oryx MLLM: On-Demand Spatial-Temporal Understanding at Arbitrary Resolution"
_ICLR.cc/2025/Conference — ICLR 2025 Poster_

### Official Review · Reviewer_qcfr · 2024-10-17

**Soundness:** 2
**Presentation:** 3
**Contribution:** 2
**Rating:** 6
**Confidence:** 4

**Summary:**

The primary goal of this work is to address the limitations of current multi-modal LLMs in processing visual inputs of varying lengths. To tackle this issue, the paper proposes two key improvements based on existing architectures: First, it enhances the current visual encoder to accept arbitrary inputs by enlarging and rescaling the positional encoding. Second, it implements input downsampling based on the type of visual input, supplemented by additional attention mechanisms to compensate for information loss. To ensure broader applicability, this work also carefully curates diverse datasets to enhance the model's capabilities. The results demonstrate that the proposed approach shows significant advantages over similarly sized models across various benchmarks.

**Strengths:**

1. The writing is clear, engaging, and easy to understand.
2. The two proposed improvements to the existing architecture are concise, straightforward, and easily implementable.
3. The approach further enhances the performance ceiling of similarly sized models across various benchmarks, contributing positively to the advancement of the community.

**Weaknesses:**

1. The title and introduction seems somewhat overclaiming. Initially, I interpreted "ON-DEMAND" in the title as implying a dynamic adjustment of spatiotemporal compression based on input; however, the method only implements fixed downsampling based on input type, which was disappointing after reviewing the methodology.
2. The proposed improvements appear somewhat rudimentary. Both the adjustments to positional encoding and the downsampling with attention seem like baseline approaches, lacking deep exploration or inspiration for future work. While I appreciate the complexity of the experiments and the contributions to the community, there are notable deficiencies in novelty.
3. There is a lack of comparison regarding processing times for different methods when handling long videos or mixed data. Although the paper emphasizes the method's efficacy for compressing long videos, I did not find any specific comparisons regarding processing times.

**Questions:**

A discussion and comparison of existing methods for compressing visual signals would significantly enhance the paper.

---

> ### Author Response · Authors · 2024-11-23
> **Official Response to Reviewer qcfr (Part 1/2)**
>
> We thank the reviewer for the careful reading and positive, insightful feedback. We address the questions and clarify the issues accordingly as described below.
>
> ---
>
> **About the Motivation**
>
> > The title and introduction seems somewhat overclaiming. Initially, I interpreted "ON-DEMAND" in the title as implying a dynamic adjustment of spatiotemporal compression based on input; however, the method only implements fixed downsampling based on input type, which was disappointing after reviewing the methodology.
>
> Thank you for the comments.  We believe that the ability of MLLM to support **adjustable and switchable** visual representations is the core contribution of Oryx, addressing a significant challenge faced by existing MLLMs.  We use the term "on-demand" to highlight this feature from the model user's perspective, meaning that users can determine Oryx's mode to meet their specific needs.  A more automated solution could be developed on top of Oryx with additional sub-modules to adaptively decide the compression ratios, and our designs in Oryx facilitate this possibility.  However, this remains an open question, as the complexity of video content and the intricacy of user prompts may present conflicting challenges.
>
> ---
>
> **About the Contribution**
>
> > The proposed improvements appear somewhat rudimentary. Both the adjustments to positional encoding and the downsampling with attention seem like baseline approaches, lacking deep exploration or inspiration for future work. While I appreciate the complexity of the experiments and the contributions to the community, there are notable deficiencies in novelty.
>
> Thank you for your comments.  We believe the main contribution of Oryx is valuable to the research community for the following reasons:
>
> 1. To the best of our knowledge, Oryx is the first open-source MLLM that supports **native-resolution inputs** with a well-trained visual encoder specifically optimized for vision-language understanding tasks.  The OryxViT encoder, along with the strategy for supporting **visual inputs at arbitrary resolutions**, represents significant progress compared to current mainstream visual representation methods based on dynamic partitioning and conventional CLIP models (or their variants).  Extensive experiments on ablation, efficiency, and performance demonstrate the superiority of OryxViT, offering a more natural and intuitive solution for MLLM fields.  The change in positional encoding is only a minor modification within the overall framework.
>
> 2. Oryx introduces a **switchable design for processing video**, an aspect previously overlooked in MLLM research.  Through our fusing operations with a dynamic compressor, we extend the capability for switchable understanding of both short and long videos into the concept of on-demand understanding, providing a user-friendly perspective for MLLM.  The downsampling with the attention module is our technical contribution to comprehensive MLLM.  We believe our approach will inspire future work to further enhance adaptability and capability in video understanding, which still falls short of human expectations.
>
> The new designs introduced in Oryx are also recognized by other reviewers. For example, **Reviewer hrJh** comments that *"The paper introduces OryxViT and the Dynamic Compressor to handle various visual inputs with arbitrary resolutions and temporal lengths, which address a critical limitation in current MLLMs."* **Reviewer o36J** comments that *"This can benefit the community since it offers a more flexible and effective alternative."*

---

> ### Author Response · Authors · 2024-11-23
> **Official Response to Reviewer qcfr (Part 2/2)**
>
> **About Comparisons on Long Videos**
>
> > There is a lack of comparison regarding processing times for different methods when handling long videos or mixed data. Although the paper emphasizes the method's efficacy for compressing long videos, I did not find any specific comparisons regarding processing times
>
> Thank you for your suggestion.   Several attempts [1,2] have been made to address the issues with long videos.   We categorize these into **compression-based approaches** (e.g., LongVA [1]) and **direct approaches** with optimizations on forward processing (e.g., LongVILA [2]).   We compare the processing times of different models and present the results below. The speed test experiments are conducted on a 64-frame video at $384 \times 384$ resolution using one NVIDIA A100 GPU. Although our method is slightly slower than LongVA and LongVILA due to differences in the visual encoder and compressor, Oryx offers a superior functional-efficiency trade-off by supporting on-demand understanding across images, short videos, and long videos.
>
> | Model    | Model Size | VideoMME Acc. | Forward Time(s) |
> | -------- | ---------- | ------------- | --------------- |
> | Oryx     | 7B         | 58.3          | 0.027           |
> | LongVA   | 7B         | 52.6          | 0.025           |
> | LongVILA | 8B         | 50.5          | 0.024           |
>
> ---
>
> **About the Compression Designs**
>
> > A discussion and comparison of existing methods for compressing visual signals would significantly enhance the paper.
>
> Thanks for the insightful comments. We will append a discussion part about compressions in visual signals in our revised paper soon. Here we provide comparisons of compressors to support our claim in the discussion. We decompose the usage of the compressing strategy into two aspects.
>
> 1. **The overall compressing architecture.** We integrate several mainstream approaches, including direct average pooling, \(2 \times 2\) spatial convolution, Q-former, and our proposed Dynamic Compressor for comparison. The results are presented below. We reference results from VideoMME and MLVU. Our observations indicate that the proposed Dynamic Compressor outperforms traditional compressing strategies based on average pooling and spatial convolution. Additionally, we find that Q-former-based methods are not suitable for handling long visual content with fixed lengths of visual tokens. This limitation arises because the information capacity of a visual token is directly proportional to its length, making fixed lengths inadequate for more complex cases involving longer visual content.
>
> | Compressing Strategy | VideoMME | MLVU     |
> | -------------------- | -------- | -------- |
> | Average Pooling      | 54.6     | 57.5     |
> | Convolution          | 54.2     | 56.8     |
> | Q-former             | 42.7     | 35.3     |
> | Dynamic Compressor   | **55.4** | **59.3** |
>
> 2. **The compressing function in Dynamic Compressor.** We are also curious about which compressing function performs better within the dynamic compressor. We compared average pooling, DWConv, and Conv-MLP architectures.  The results indicate that the parameter-free average pooling outperforms parameter-dependent methods like DWConv and Conv-MLP.  This improvement is likely because average pooling better preserves the distribution of visual features, whereas more complex downsampling layers may not be effectively trained with the current pipeline.
>
> | Compressing Function | VideoMME | MLVU     |
> | -------------------- | -------- | -------- |
> | DWConv               | 55.0     | 58.9     |
> | Conv-MLP             | 54.7     | 58.5     |
> | Average Pooling      | **55.4** | **59.3** |

---

> ### Author Response · Authors · 2024-11-25
> **Official Comment by Authors**
>
> Dear Reviewer qcfr:
>
> We sincerely appreciate your great efforts in reviewing this paper.   Your constructive advice and insightful comments really help improve our paper.   We have carefully tried to address your concerns with detailed explanations, essential experiments, and the revised version of our manuscripts.
>
> Considering the approaching deadline, we would appreciate it if you could spare some time to go over our response, and we can further address unclear explanations and remaining concerns if any.   We sincerely hope that you can consider reevaluating the initial assessment if we have successfully addressed your concerns.
>
> Best Regards,
>
> The Authors

---

> > ### Comment · Reviewer_qcfr · 2024-11-26
> >
> > Thanks for the author's response, which has resolved most of the concerns. I am willing to increase my score.

---

### Official Review · Reviewer_o36J · 2024-11-03

**Soundness:** 3
**Presentation:** 3
**Contribution:** 3
**Rating:** 6
**Confidence:** 4

**Summary:**

This paper proposes a series of improvement on the vision encoder design for multi-modal large language models, enabling better understanding of images, videos, and 3D multi-view inputs. The core contribution is a ViT-based vision encoder that supports native resolution inputs, accommodating varying resolutions and aspect ratios. Additionally, the authors introduce a dynamic compression technique with a cross-attention design to reduce the information loss caused by the pooling operations. The results demonstrate the model's effectiveness across video, 3D, and image understanding tasks.

**Strengths:**

1. This paper implements and open-sources a powerful ViT-based encoder that supports native input resolution. This can benefits the community since it offers a more flexible and effective alternative to dynamic partitioning or fixed resolution approaches.
2. The reported results are very promising, especially on tasks requires high-resolution details or long context understanding.

**Weaknesses:**

1. The main concern is the lack of detailed implementation regarding the pretraining of the Oyrx ViT. It would be valuable to understand whether the Oyrx ViT remains effective without pretraining—for instance, by modifying the ViT architecture as Oyrx ViT and directly fine-tuning it on multi-modal conversation data. Additionally, more specifics about the pretraining process, such as the dataset size, batch size, and training objectives, should be provided.
2. The cross-attention downsampling module is similar to prior work [1]. The authors should discuss how their approach differs from the cross-attention architecture proposed by mini-Gemini[1].  It seems that the only apparent difference is that mini-Gemini uses a CLIP encoder for low-resolution features.
[1] Mini-Gemini: Mining the Potential of Multi-modality Vision Language Models https://arxiv.org/abs/2403.18814

**Questions:**

1. What is the performance of Oyrx ViT without pretraining? If pretraining is essential, could the authors provide details on the amount of data used and the specific task mixture involved in the pretraining process?
2. In section 3.1.3, how to utilize the correspondence information provided by TrackingAnything?

---

> ### Author Response · Authors · 2024-11-23
> **Official Response to Reviewer o36J (Part 1/2)**
>
> We thank the reviewer for the careful reading and positive, insightful feedback. We address the questions and clarify the issues accordingly as described below.
>
> ---
>
> **About the Details for OryxViT**
>
> >  The main concern is the lack of detailed implementation regarding the pretraining of the Oyrx ViT. It would be valuable to understand whether the Oyrx ViT remains effective without pretraining—for instance, by modifying the ViT architecture as Oyrx ViT and directly fine-tuning it on multi-modal conversation data. If pretraining is essential, could the authors provide details on the amount of data used and the specific task mixture involved in the pretraining process? Additionally, more specifics about the pretraining process, such as the dataset size, batch size, and training objectives, should be provided.
>
> Thank you for the comments. We provide details about OryxViT here and will include relevant information in our revised appendix. We pre-train OryxViT with a relatively small language model (Qwen2-0.5B in our implementation) to enhance the language interface and improve vision-language alignment. We unfreeze OryxViT and apply LoRA fine-tuning to the language models. As a result, the total number of trainable parameters is 0.6B, making the training process significantly faster than supervised fine-tuning in the main stage (approximately **10 times faster**). We collected a total of 400M pre-training data, focusing mainly on image captioning and image OCR tasks. For image captioning, we used the CapsFusion datasets, and for OCR tasks, we employed synthesized OCR data pairs with OCR models. We set the batch size to 2048 and used a similar cross-entropy loss as in the main stages.
>
> We are also interested in the actual effectiveness of the arbitrary-resolution design.  To analyze this, we conducted experiments and list the results below.  We found that directly using the SigLIP architecture with native-resolution inputs and proceeding immediately to the supervised fine-tuning stage results in poor performance, as the model is not well-suited for the task.  However, using a **small amount of pre-training data (10M)** at arbitrary resolutions allows the model to initialize quickly, achieving strong performance on multi-modal benchmarks.  Training with more data can further improve the scores slightly.  Therefore, we emphasize the contribution of OryxViT with native-resolution inputs.
>
> | Visual Encoder | Resolution        | Training Samples | OCRBench | MMBench |
> | -------------- | ----------------- | ---------------- | -------- | ------- |
> | OryxViT        | Native Resolution | 0 (SigLIP)       | 67       | 15.8    |
> | OryxViT        | Native Resolution | 10M (Subset)     | 557      | 68.7    |
> | OryxViT        | Native Resolution | 400M             | 572      | 69.3    |

---

> ### Author Response · Authors · 2024-11-23
> **Official Response to Reviewer o36J (Part 2/2)**
>
> **About the Downsampling Layer**
>
> > The cross-attention downsampling module is similar to prior work [1]. The authors should discuss how their approach differs from the cross-attention architecture proposed by mini-Gemini[1]. It seems that the only apparent difference is that mini-Gemini uses a CLIP encoder for low-resolution features. [1] Mini-Gemini: Mining the Potential of Multi-modality Vision Language Models https://arxiv.org/abs/2403.18814
>
> The Dynamic Compressor module has a different **motivation** and **architecture** compared to the architecture in Mini-Gemini [1].
>
> 1. **Motivation.** The goal of Mini-Gemini is to enhance low-resolution features with information from high-resolution features, serving as an alternative to providing high-resolution visual inputs.  This approach shares a similar objective with dynamic partitioning or native-resolution inputs.  In contrast, the Dynamic Compressor aims to support a switchable design for both full and downsampled features, which is crucial for enabling on-demand choices with support for both high and low compression ratios.
>
> 2. **Architecture.** Mini-Gemini is primarily motivated by data mining and fusion, employing a global attention operation.  On the other hand, the Dynamic Compressor focuses on downsampling, utilizing regional attention within the downsampled regions.  We do not use a dual encoder implementation;  instead, the downsampled features are derived from the original features, ensuring that the multi-granularity features remain valid for visual tokens with similar distributions.  However, in Mini-Gemini, only the low-resolution features are used for understanding.
>
> The similarity between the Dynamic Compressor and Mini-Gemini lies in the cross-attention approach, which effectively aids in compression.  Our inspiration comes from previous work on visual compression including BLIP-2 [2] and Mini-Gemini [1].
>
> ---
>
> **About the Correspondence**
>
> > In section 3.1.3, how to utilize the correspondence information provided by TrackingAnything?
>
> We start by using TrackingAnything to track the primary objects across multiple views. Next, we label identical objects across different frames. The model is then tasked with answering questions, utilizing the information that identical labels across frames correspond to the same object. We encourage the model to use this information by incorporating it into the system prompt. This approach enhances the model's ability to establish a comprehensive 3D spatial relationship based on these coarse correspondence labels. The observed improvement on the ScanQA dataset demonstrates the effectiveness of our method.
>
> ---
>
> **References**
>
> [1] Yanwei Li, et, al. Mini-Gemini: Mining the Potential of Multi-modality Vision Language Models
>
> [2] Junnan Li, et, al. BLIP-2: Bootstrapping language-image pre-training with frozen image encoders and large language models

---

> > ### Comment · Reviewer_o36J · 2024-11-28
> >
> > Thanks for the authors for the detailed feedback. I remain my positive rating of the paper after the rebuttal.

---

> ### Author Response · Authors · 2024-11-25
> **Official Comment by Authors**
>
> Dear Reviewer o36J:
>
> We sincerely appreciate your great efforts in reviewing this paper.   Your constructive advice and insightful comments really help improve our paper.   We have carefully tried to address your concerns with detailed explanations, essential experiments, and the revised version of our manuscripts.
>
> Considering the approaching deadline, we would appreciate it if you could spare some time to go over our response, and we can further address unclear explanations and remaining concerns if any.   We sincerely hope that you can consider reevaluating the initial assessment if we have successfully addressed your concerns.
>
> Best Regards,
>
> The Authors

---

### Official Review · Reviewer_hrJh · 2024-11-04

**Soundness:** 3
**Presentation:** 3
**Contribution:** 3
**Rating:** 6
**Confidence:** 3

**Summary:**

This paper introduces Oryx, a unified multimodal large language model (MLLM) designed for comprehensive spatial-temporal understanding across diverse visual inputs, including images, videos, and multi-view 3D scenes.

Key innovations include the OryxViT encoder, which processes inputs with variable aspect ratios and resolutions, and a dynamic compressor that supports on-demand compression ratios from 1x to 16x based on task requirements. It allows the model to handle extended temporal lengths while maintaining performance.

The paper also highlights enhanced data curation and specialized training strategies that bolster Oryx’s capabilities in image, video, and 3D multimodal understanding. Comprehensive evaluations across multiple benchmarks demonstrate Oryx’s competitive performance, often surpassing larger models in specific tasks.

**Strengths:**

1. **Innovative Architecture**:
The paper introduce OryxViT and the Dynamic Compressor to handle various visual inputs with arbitrary resolutions and temporal lengths, which address a critical limitation in current MLLMs.

2. **Support for Native Resolution**:
The OryxViT encoder preserves the integrity of visual content by processing inputs at their original resolution, which is essential for detailed tasks.

3. **Efficiency and Scalability**:
By supporting on-demand compression, Oryx can effectively manages computational resources. It’s suitable for processing both high-resolution images and long-duration videos without compromising performance.

4. **Comprehensive Evaluation**:
The model is extensively tested across a wide range of benchmarks, including NextQA, Perception Test, MMBench-Video, and ScanQA. Demonstrate its robust performance and state-of-the-art results among open-source models.

5. **Advanced Training Strategies**:
The paper also introduce an effective data curation and training pipelines, including long-form temporal training and spatial-aware knowledge acquisition through coarse correspondences to boost MLLM training.

**Weaknesses:**

1. **Writing Issues**:
    - **Lack of Focus:** The first part of the paper mainly focus on addressing the varying resolutions of visual inputs, while the second part shifts to enhanced data curation for supporting different modalities (e.g. 3D scene understanding) in MLLMs, which create a sense of discontinuity.
    - **Excessive Technical Detail**: While the technical innovations are commendable, the method part contains too much technical details that can disrupt the flow and make it difficult to read.
2. **Ambiguity in Performance Gain:**
    - It also remains unknown whether the observed performance improvements on Long-form temporal/3D understanding mainly come from the model design or from the introduction of the new long-form videos dataset and the Coarse Correspondences dataset. A more detailed analysis is required.
3. **More experiment to justify design choice:**
    - There are too many manual design choices within this paper that haven’t been justified by exp.

        1. A more detailed analysis of the computational complexity of the Variable-Length Self-Attention approach compared to traditional methods would be helpful. How does the memory footprint scale with varying sequence lengths, and what optimization strategies are in place to handle potential memory bottlenecks?

        2. How does the significant reduction in token length for long videos (downsampling by a factor of 16) affect the model's ability to capture fine-grained visual details essential for understanding complex scenes or actions? Why downsample (1x, 4x, 16x) is a reasonable choice? How would other downsample strategy affect performance?

        3. Does using a shared MLP for projecting features from different downsampling ratios adequately capture the unique characteristics of images, short videos, and long videos, or could specialized expert projection layers yield better performance?
4. **Failure cases analysis**
    - Including analysis of failure cases would be appreciated. This analysis could provide valuable insights into the limitations of the Oryx model and offer transparency on how it performs under various challenging scenarios.

**Questions:**

Given the current weaknesses of this paper, I am unable to provide a positive score. However, addressing the issues above could prompt me to reconsider the rating.

---

> ### Author Response · Authors · 2024-11-23
> **Official Response to Reviewer hrJh (Part 1/5)**
>
> We thank the reviewer for the careful reading and positive, insightful feedback. We address the questions and clarify the issues accordingly as described below.
>
> ---
>
> **About the Writing Issues**
>
> Thank you for your comments and detailed review of our manuscript. We have carefully revised the paper accordingly and have provided our responses below. Please refer to the revised method section in the newly uploaded PDF file. The changed part is highlighted in blue color.
>
> ---
>
> > Lack of Focus: The first part of the paper mainly focus on addressing the varying resolutions of visual inputs, while the second part shifts to enhanced data curation for supporting different modalities (e.g. 3D scene understanding) in MLLMs, which create a sense of discontinuity.
>
> We want to clarify that our goal is to develop Oryx as a competitive VLM system for on-demand understanding.  To achieve this, we have focused on improvements in both architecture design—specifically, the arbitrary-resolution visual encoder and on-demand dynamic compressor—and data preparation, including specialized training with long videos and corresponding 3D data.  Both the architecture and data components are integral to the manuscript and should be included in the methods section.
>
> We acknowledge that the current structure might make it a bit difficult to fully understand details of our approach, as we are enhancing MLLM from multiple angles.  We believe the following changes could help readers better understand our paper:
>
> Section 3.1.3 (One Model for All: Image, Video, and 3D Understanding) currently emphasizes data preparation, which could be better introduced in Section 3.2.  This way, Section 3.1 will focus solely on architecture design, while Section 3.2 will cover our efforts in the training pipeline and data integration, providing clearer insights.  We welcome further suggestions!
>
> ---
>
> > Excessive Technical Detail: While the technical innovations are commendable, the method part contains too much technical details that can disrupt the flow and make it difficult to read.
>
> Thanks for the comments. We have proofread the whole method part about the contents of technical details. We have adjust the flow and the representation accrodingly. You can refer to our newly uploaded PDF file.
>
> ---
>
> **About the Performance Gain**
>
> > It also remains unknown whether the observed performance improvements on Long-form temporal/3D understanding mainly come from the model design or from the introduction of the new long-form videos dataset and the Coarse Correspondences dataset. A more detailed analysis is required.
>
> Thank you for your comments.  As previously stated, both the **architecture design** and **data curation** are core contributions of our manuscript.  We have presented the analysis related to the architecture design in Table 5(a).  Additionally, we have included comprehensive ablation studies on the architecture and data preparation here for clearer analysis.  Please note that the ablation experiments were conducted on a subset of the training datasets.  The results indicate that the improved architecture design primarily enhances performance on the general video understanding benchmark, while the long-form temporal data benefits the long video benchmarks. We want to emphasize that our new designs (i.e., the new ViT and the dynamic Compressor) are intended not only to improve (or at least maintain) performance but also to enable the on-demand features of MLLMs.  We will include these ablations in the final version of the paper soon.
>
> | Connector          | Long-Form Data | VideoMME | MLVU     |
> | ------------------ | -------------- | -------- | -------- |
> | MLP                | &#10007;       | 54.5     | 56.4     |
> | Dynamic Compressor | &#10007;       | 55.2     | 58.1     |
> | Dynamic Compressor | &#10004;       | **55.4** | **59.3** |
>
> For 3D-related tasks, our proposed approach based on coarse correspondence is an effective solution for 3D benchmarks. We conducted experiments and provided results on ScanQA. **"3D Data"** indicates whether 3D datasets are included in the training set, while **"Coarse Correspondence"** refers to using the approach to annotate objects in 3D-relevant videos. We observe that both components contribute to improved understanding results. We will include these ablations in the final version of the paper soon.
>
> | 3D Data  | Coarse Correspondence | METEOR   | ROUHE-L  |
> | -------- | --------------------- | -------- | -------- |
> | &#10007; | &#10007;              | 11.7     | 28.1     |
> | &#10004; | &#10007;              | 12.8     | 32.7     |
> | &#10004; | &#10004;              | **14.0** | **35.1** |

---

> ### Author Response · Authors · 2024-11-23
> **Official Response to Reviewer hrJh (Part 2/5)**
>
> **About the Design Choice**
>
> Thank you for your insightful suggestion. We completely agree that the experiments listed below are essential to demonstrate the effectiveness of the Oryx model. We have carefully evaluated and verified these variants in our early experiments. In the following part, we have re-evaluated the ablations mentioned above and provided the corresponding experimental results.
>
> ---
>
> > A more detailed analysis of the computational complexity of the Variable-Length Self-Attention approach compared to traditional methods would be helpful. How does the memory footprint scale with varying sequence lengths, and what optimization strategies are in place to handle potential memory bottlenecks?
>
> The primary motivation for using Variable-Length Self-Attention is to manage the variable visual token lengths and process the vision transformer in batches, thereby avoiding "for" loops in the implementation, which significantly impact the visual encoder's throughput.  By utilizing *FlashAttention* in the implementation of Variable-Length Self-Attention, the memory overhead and inference throughput remain negligible, as the variable-length attention operation is fully optimized through modifications in the CUDA kernels.
>
> Moreover, the model size of the Vision Transformer is considerably smaller than that of large language models (400M parameters compared to 7B/32B).  Consequently, the main memory cost arises from the weights and features of the LLMs, as full attention is computed on visual tokens within the LLM, even when using dynamic partitioning in visual encoding.  Therefore, we maintain similar efficiency to previous solutions in terms of inference speed and memory cost.
>
> Below are our experimental results regarding memory costs. We conducted the inference procedure on an NVIDIA A100 GPU, setting the batch size to 4 and the image size to a total of $1280\times 1280$ pixels, with the aspect ratio randomly determined.  The dynamic-partition baseline uses an average of **48.7G** of memory, while OryxViT uses **49.1G**, showing comparable results. We will include these results in the final version of the paper soon.
>
> | Backbone | Divide Approach   | Max Memory Cost |
> | -------- | ----------------- | --------------- |
> | OryxViT  | Native Resolution | 49.1GB          |
> | SigLIP   | Dynamic Partition | 48.7GB          |
>
> Additionally, we plotted the **memory-resolution curve** to illustrate how the memory footprint increases with varying resolutions.    Our experiment was conducted on an NVIDIA A100 GPU, using square-shaped input images.    The results show that the memory cost is positively correlated with image resolution and is **approximately in linear relation** among common image resolutions, indicating that OryxViT provides a memory-efficient solution when scaling input resolutions.
>
> Furthermore, we can also observe that the primary memory cost arises from the **model weights and the features themselves**.   Notably, considering that the main memory cost comes from the computation in LLMs, our method with native resolution support will not introduce much extra memory footprint compared to previous methods like LLaVA-Next and InternVL2.   Oryx can largely save memory if we apply dynamic compression on visual tokens (e.g., 2x or 4x downsampling) while previous methods do not support this feature. We will include these results in the final version of the paper soon.
>
> Figure link: https://drive.google.com/file/d/16NMD6FeMu_f2vGYg-QVwIMjU2zcofPok/view?usp=sharing (Anonymous Link)

---

> ### Author Response · Authors · 2024-11-23
> **Official Response to Reviewer hrJh (Part 3/5)**
>
> > How does the significant reduction in token length for long videos (downsampling by a factor of 16) affect the model's ability to capture fine-grained visual details essential for understanding complex scenes or actions?
>
> The number of video frames and the compression ratios serve as a trade-off given the maximum token length, which is limited by GPU memory.  For **short videos**, we maintain a relatively low compression ratio to preserve the model's ability to capture complex and detailed information.  In contrast, for **extremely long videos**, sparse frame sampling may result in the loss of critical information in certain parts.  Therefore, we need more frames, and a decrease in the quality of each frame is acceptable.  Our goal is to bridge the gap between high and low compression ratios by proposing an on-demand choice for users, offering a switchable model with strong performance in both modes through our method designs.
>
> We evaluated our approach on the VideoMME benchmark, maintaining the same overall number of tokens while using different compression ratios.  We provide results for 4x downsampling with 64 frames and 16x downsampling with 256 frames.  The results show that a lower compression ratio performs better on short and medium-length videos, while a higher compression ratio is more effective for long videos.  This indicates that although a higher downsampling ratio affects the model's ability to process each frame, providing denser information over time improves performance on long video benchmarks.  We integrate the 4x downsampling ratio in our experiments for better overall performance, as most benchmarks focus on knowledge within certain frames.
>
> | Downsample Ratio | Frame Number | Overall Acc. | Short Acc. | Medium Acc. | Long Acc. |
> | ---------------- | ------------ | ------------ | ---------- | ----------- | --------- |
> | 4x               | 64           | **59.8**     | **71.0**   | **56.3**    | 49.6      |
> | 16x              | 256          | 58.6         | 69.6       | 55.2        | **50.0**  |
>
> ---
>
> > Why downsample (1x, 4x, 16x) is a reasonable choice?
>
> We selected **three representative modes** for downsampling based on the following considerations:
>
> 1. We maintain 1x downsampling for images to fully preserve image understanding performance and to keep settings consistent with the previous stage.
>
> 2. The 4x downsampling is derived from 2x2 structural downsampling, a common and natural ratio for spatial reduction.
>
> 3. We also choose 16x downsampling, developed from 4x4 structural downsampling, to achieve a high compression ratio suitable for extremely long videos. This allows us to support input of 1600 frames with a total length of 32k for LLM.
>
> We believe that supporting fine-grained and adaptive downsampling ratios with more options could be a valuable direction for future research.

---

> ### Author Response · Authors · 2024-11-23
> **Official Response to Reviewer hrJh (Part 4/5)**
>
> > How would other downsample strategy affect performance?
>
> We decompose the usage of the downsampling strategy into two aspects.
>
> 1. **The overall downsampling architecture.** We integrate several mainstream approaches, including direct average pooling, $2 \times 2$ spatial convolution, Q-former, and our proposed Dynamic Compressor for comparison. The results are presented below, and we will include these ablations in the final version of the paper soon. We reference results from VideoMME and MLVU. Our observations indicate that the proposed Dynamic Compressor outperforms traditional downsampling strategies based on average pooling and spatial convolution. Additionally, we find that Q-former-based methods are not suitable for handling long visual content with fixed lengths of visual tokens. This limitation arises because the information capacity of a visual token is directly proportional to its length, making fixed lengths inadequate for more complex cases involving longer visual content.
>
> | Downsamling Strategy | VideoMME | MLVU     |
> | -------------------- | -------- | -------- |
> | Average Pooling      | 54.6     | 57.5     |
> | Convolution          | 54.2     | 56.8     |
> | Q-former             | 42.7     | 35.3     |
> | Dynamic Compressor   | **55.4** | **59.3** |
>
> 2. **The downsampling function in Dynamic Compressor.** We are also curious about which downsampling function performs better within the dynamic compressor. We compared average pooling, DWConv, and Conv-MLP architectures. The results are presented below, and we will include these ablations in the final version of the paper soon. Results indicate that the parameter-free average pooling outperforms parameter-dependent methods like DWConv and Conv-MLP.  This improvement is likely because average pooling better preserves the distribution of visual features, whereas more complex downsampling layers may not be effectively trained with the current pipeline.
>
> | Downsamling Function | VideoMME | MLVU     |
> | -------------------- | -------- | -------- |
> | DWConv               | 55.0     | 58.9     |
> | Conv-MLP             | 54.7     | 58.5     |
> | Average Pooling      | **55.4** | **59.3** |
>
> ---
>
> > Does using a shared MLP for projecting features from different downsampling ratios adequately capture the unique characteristics of images, short videos, and long videos, or could specialized expert projection layers yield better performance?
>
> We explored using specialized projection layers when integrating video modality into our model.  However, we found that employing a shared MLP for all visual inputs yields better performance.  To align with this design, we use the Dynamic Compressor module to maintain the distribution of image and video features closely, and employ the shared MLP for visual information fusion.  This ensures that input tokens for LLMs maintain a consistent distribution for both images and videos, allowing the shared MLP to be better trained with joint visual knowledge.
>
> We conducted experiments to support our hypothesis, comparing our shared MLP strategy with separate MLPs for images and videos.  Both the image and video MLPs were initialized from pre-trained image weights.  The results indicate that using separate MLPs **negatively impacts** video benchmarks, as the dual-projector design can lead to differing distributions for similar data.  Therefore, using a single MLP is a more effective solution for visual encoding, a finding also supported by concurrent work [1]. We will add the results in our final version of the paper soon.
>
> | MLP       | VideoMME | MLVU | MMBench | MMMU |
> | --------- | -------- | ---- | ------- | ---- |
> | Shared    | 55.4     | 59.3 | 81.4    | 43.9 |
> | Separated | 54.0     | 54.2 | 81.2    | 43.1 |
>
> ---
>
> **References**
>
> [1] Bo Li, et al. LLaVA-OneVision: Easy Visual Task Transfer

---

> ### Author Response · Authors · 2024-11-23
> **Official Response to Reviewer hrJh (Part 5/5)**
>
> **About the Failure Cases**
>
> > Including analysis of failure cases would be appreciated. This analysis could provide valuable insights into the limitations of the Oryx model and offer transparency on how it performs under various challenging scenarios.
>
> Thanks for the suggestion. We have further tested the Oryx model on more challenging samples. We provide some representative failure cases to show the limitations of the Oryx model and point out the future direction for VLMs. We will add relevant cases in our revised appendix material soon.
>
> ---
>
> **Failure Case 1: Time Perception**
>
> ***Video:** https://streamable.com/rubio9 (Anonymous Link)*
>
> ***Question:** Describe what happened at 15-18 seconds. Describe the scene in detail.*
>
> ***Oryx:** At 15-18 seconds, the video captures a moment during the 2018 Wimbledon Semifinal between Novak Djokovic and Rafael Nadal. The scene is set on a grass court with players positioned on opposite sides of the net. The scoreboard in the background shows that the match is tied at 7-7, indicating a closely contested game. The player on the left side of the court is preparing to hit the ball, while his opponent on the right side is anticipating the shot. The audience in the stands is visible, adding to the atmosphere of the event. (The answer is wrong. in 15-18 seconds, djokovic wins the point with a forehand shot.)*
>
> From the responses, we can observe that current video understanding models using uniform frame sampling tend to lose the temporal context in videos, which is critical information between frames. For instance, Oryx provides incorrect answers and cannot accurately determine the timestamps for the given video. We believe that incorporating timestamps directly into the video or processing video inputs without frame sampling may help address this issue, leading to a more comprehensive video understanding capability.
>
> ---
>
> **Failure Case 2: Knowledge Reasoning and Transfer**
>
> ***Video:** https://streamable.com/l51acc (Anonymous Link)*
>
> ***Question:** What is the purpose of beating the balloon? Describe in detail.*
>
> ***Oryx:** The purpose of beating the balloon is to create a series of air pockets within it. This process involves repeatedly striking the balloon with a flat, blunt object, such as a wooden spoon or a similar tool, until it becomes visibly deflated and has multiple air pockets throughout its surface. The air pockets are created by the impact of the object on the balloon's surface, causing the air inside to be released and trapped in the form of bubbles. (The answer is wrong. The correct purpose is to conduct chemical reaction and generate the cute globefish. )*
>
> From the responses, we can observe that the Oryx model lacks knowledge of chemical reactions, indicating that the training data of current MLLMs is not comprehensive enough. We believe that expanding the knowledge base of current MLLMs is an urgent issue. Additionally, this example requires simple reasoning to integrate the chemical knowledge into the main idea of the video, which poses a challenge for Oryx. Enhancing the reasoning capability may help address this issue.

---

> ### Author Response · Authors · 2024-11-25
> **Official Comment by Authors**
>
> Dear Reviewer hrJh:
>
> We sincerely appreciate your great efforts in reviewing this paper.   Your constructive advice and insightful comments really help improve our paper.   We have carefully tried to address your concerns with detailed explanations, essential experiments, and the revised version of our manuscripts.
>
> Considering the approaching deadline, we would appreciate it if you could spare some time to go over our response, and we can further address unclear explanations and remaining concerns if any.   We sincerely hope that you can consider reevaluating the initial assessment if we have successfully addressed your concerns.
>
> Best Regards,
>
> The Authors

---

> > ### Comment · Reviewer_hrJh · 2024-11-26
> > **Official Comment by Reviewer hrJh**
> >
> > I appreciate the detailed response and effort from the authors. The experiments addressed most of my questions, and I believe the paper's soundness and contributions are sufficient for acceptance. As a result, I have raised my score.

---

### Official Review · Reviewer_ceNe · 2024-11-09

**Soundness:** 3
**Presentation:** 3
**Contribution:** 3
**Rating:** 6
**Confidence:** 3

**Summary:**

Oryx MLLM paper proposes unified solution to process images, spatio-temporal videos and multi-view 3D input. It introduces a dynamic compressor module that performs token compression and adaptive positional embedding enables native resolution input / video length.

**Strengths:**

- The one-fit all solution that unifies solution to video of any length and image of varying resolution, 3D input makes is a significant contribution in it's usability especially in real world deployment.
-The code/ training details have been elaborated and released which enables reproducibility and usage.
- Exhaustive benchmarks provided for the long video understanding and the 3D correspondance understanding.

**Weaknesses:**

- The effect of ORyxViT and the compressor module on the inference throughput needs to be provided.
- Need clarity on what is short and long video to decide the downsample layers: Line 223
- It would be great to see apple-apple comparison of SOTA VLM/ Video-LLM outputs for the qualitative examples that were added.

**Questions:**

- Covered above in weakness.

---

> ### Author Response · Authors · 2024-11-23
> **Official Response to Reviewer ceNe (Part 1/3)**
>
> We thank the reviewer for the careful reading and positive, insightful feedback. We address the questions and clarify the issues accordingly as described below.
>
> ---
>
> **About the Inference Speed**
>
> > The effect of OryxViT and the compressor module on the inference throughput needs to be provided.
>
> Thank you for your suggestion. As mentioned in lines 203-211 of the manuscript, we implement variable-length self-attention using the highly optimized *FlashAttention* library. This allows the inference throughput of our arbitrary-resolution visual encoder to remain comparable to the dynamic partition approach used in previous methods like LLaVA [1] and InternVL2 [2].
>
> We tested the inference speed with an input image size of $1280 \times 1280$ on one NVIDIA A100 GPU, and the results are shown below. We observed that OryxViT is only **7%** slower than SigLIP with the dynamic partition approach. We believe this overhead is acceptable given the improved performance and the ability to process images at their native resolutions directly. We will include these results in the final version of the paper soon.
>
> | Backbone | Processing Approach | Throughput (images/s) |
> | -------- | ------------------- | --------------------- |
> | OryxViT  | Native Resolution   | 146.5                 |
> | SigLIP   | Dynamic Partition   | 157.7                 |
>
> For the **compressor module**, we use a lightweight attention block to reduce computational costs (with a *Softmax* projection dimension of 256, compared to an LLM hidden dimension of 3584). Additionally, the main computation for MLLM during inference occurs in the LLM forward process, making the module's overhead negligible compared to an MLP-only architecture.
>
> We compare the inference speed among the Dynamic Compressor, the MLP baseline, and MLP with average pooling, presenting the results below. Experiments are conducted on a 64-frame video at $384 \times 384$ resolution using one NVIDIA A100 GPU. We measure the total forward time for the MLLM. The results demonstrate that the downsampling operation significantly accelerates inference speed, while the dynamic compressor and average downsampling exhibit similar inference speeds. We will include these results in the final version of the paper soon.
>
> | Compressor            | Downsample Ratio | Forward Time (s) |
> | --------------------- | ---------------- | ---------------- |
> | MLP                   | 1x               | 0.053            |
> | Average Pooling + MLP | 4x               | 0.025            |
> | Dynamic Compressor    | 4x               | 0.027            |
>
> ---
>
> **About the short / long videos**
>
> > Need clarity on what is short and long video to decide the downsample layers: Line 223
>
> Thank you for the comments.  In practice, during inference, the distinction between short and long videos is determined by the user's command, aligning with our goal of on-demand understanding.  When evaluating Oryx on video benchmarks, we use a higher downsampling ratio if the video tokens exceed the context length of our LLM (i.e., 16,384).  For training, we propose a straightforward solution to learn representations of both short and long videos simultaneously.  We categorize the visual inputs based on the dataset source, applying the short video setting for general video datasets and the long video setting for datasets like "Video Needle-in-a-Haystack" with MovieNet.
>
> ---
>
> **References**
>
> [1] Haotian Liu, et, al. LLaVA-Next: Improved reasoning, ocr, and world knowledge
>
> [2] OpenGVLab Team. InternVL2: Better than the Best—Expanding Performance Boundaries of Open-Source Multimodal Models with the Progressive Scaling Strategy

---

> ### Author Response · Authors · 2024-11-23
> **Official Response to Reviewer ceNe (Part 2/3)**
>
> **About the qualitative results**
>
> > It would be great to see apple-apple comparison of SOTA VLM/ Video-LLM outputs for the qualitative examples that were added.
>
> Thank you for the insightful comments. We use 1) the state-of-the-art vision-language model, LLaVA-OneVision-7B [1], and 2) the representative long-video language model, LongVILA [2], for apple-to-apple comparisons on the samples in the appendix. The results and analysis are shown below. We will include these comparisons after the results for the Oryx model in our appendix soon. Key information in the generated results is **bolded**, and incorrect results are ~~strikeouted~~.
>
> ---
>
> **Example 1:** **Video Summarization and Detailed Description (Figure 4 in the Appendix)**
>
> ***Oryx:** The video captures a moment from the 2018 Wimbledon Semifinal between **Djokovic and Nadal**. The scene is set on a grass court, with the players positioned on opposite sides of the net. Djokovic, dressed in white, is seen preparing to **hit the ball**, while Nadal, also in white, is poised to **return the shot**. The scoreboard indicates that the match **is tied at 7-7**, adding to the tension of the moment. ....*
>
> ***LLaVA-OneVision:** The video begins with a wide shot of a grass tennis court during the 2018 Wimbledon Semifinal. Two players, both dressed in white, are positioned on either side of the net. The player on the left is preparing to serve, while the player on the right is ready to return. The court is surrounded by a large audience, and the scoreboard in the background shows the names ~~'SACHSNOVIC' and 'DRAKOVIC'~~ with scores. The player on the left serves the ball, and the player on the right returns it with a backhand. ...*
>
> ***LongVILA:** The video showcases a series of intense tennis matches at the 2018 Wimbledon Semifinal. It begins with a match **between Djokovic and Nadal**, where the scoreboard indicates that Djokovic ~~is leading 2 sets to 1~~. The players are seen serving, returning, and volleying the ball across the net on a grass court. The audience is visible in the background, and the umpire stands behind the net overseeing the match. ...*
>
> For the description tasks, Oryx produces more accurate results about the match information, the name, and the status of the player. In contrast, LLaVA-OneVision shows the wrong name, and LongVILA tells the wrong score on the board.
>
> ---
>
> **Example 2: Video Multiple Choice and Reasoning (Figure 5 in the Appendix)**
>
> ***Oryx:** **The answer is B**, because expired aspirin is a kind of drug, drugs are toxic waste. The video explains that if you have some expired aspirin, it is considered toxic waste. This is because drugs can be harmful to the environment and should not be disposed of in the regular trash. Instead, they should be taken back to a pharmacy or other designated location for proper disposal.*
>
> ***LLaVA-OneVision:** Expired aspirin is categorized as toxic waste, as indicated in the video. Toxic waste includes substances like medicines, which need special care when disposing of them to prevent environmental and health risks. **Therefore, the correct answer is B.** Toxic Waste. It's important to dispose of such items properly, often through designated hazardous waste disposal facilities or special collection events.*
>
> ***LongVILA:** **B. Toxic Waste.** The video explains that toxic waste includes medicines, batteries, dried paint, old bulbs, and dried shoe polish. Expired aspirin would fall under this category as it is a medicine.*
>
> In reasoning tasks, all the models perform the true answer. In detail, Oryx and LLaVA-OneVision show better reasoning chains than LongVILA.

---

> ### Author Response · Authors · 2024-11-23
> **Official Response to Reviewer ceNe (Part 3/3)**
>
> **Example 3: Skill Learning From Videos (Figure 6 in the Appendix)**
>
> ***Oryx:** To cite the paper titled 'Oryx', you should follow these steps:*
>
> 1. *Open google chrome and search for google scholar.*
>
> 2. *Search for the paper 'Oryx’.*
> 3. *Select the paper from the search results and click on the "Cite" button to generate the citation in your preferred format.*
>
> 4. *Copy the generated citation and paste it into your document where you want to reference the paper.*
>
> ***LLaVA-OneVision：*** *To cite the paper titled 'Oryx' from the video, follow these steps: Open the Google Scholar webpage and search for the paper 'Oryx'. Once you find the paper, ~~click on the three dots next to it~~ and select 'Cite'. Choose the citation format you need, ~~such as APA, MLA, or Chicago~~. If the paper's details are not automatically filled, ~~manually input the information from the paper's publication details including authors, title, year, and journal~~. Review the citation to ensure all information is accurate. Copy the citation and paste it into your bibliography or reference list. By following these steps, you can properly cite the paper 'Oryx' in your academic work.*
>
> ***LongVILA:** 1. Open Google Scholar and search for the paper titled 'Oryx'. 2. Click on the paper to open its details page. 3. Click on the 'Cite' button. 4. Select the citation style from the dropdown menu. 5. Copy the citation text.*
>
> In the skill-learning tasks related to citing a specific paper, the instructions provided by LongVILA are brief and lack key explanations. In contrast, Oryx and LLaVA-OneVision offer more practical steps. Although LLaVA-OneVision provides additional instructions, some detailed steps are not depicted in the original video. We believe this hallucination may stem from the information in the training data.
>
> ---
>
> **Example 4: Understanding 3D with Coarse Correspondences (Figure 7 in the Appendix)**
>
> ***Oryx:** If Alice comes out of the building and wants to go down the stairs and cross the street using the pedestrian crossing, she should **turn right after exiting the door**. This is because the pedestrian crossing is located on the right side of the entrance when facing it from inside the building. The labels inside the images indicate that the pedestrian crossing (label 3) is to the right of the door (label 4), and the stairs (label 1) are also situated to the right of the door when viewed from Alice's perspective. Therefore, turning right would lead Alice towards the pedestrian crossing and then across the street.*
>
> ***LLaVA-OneVision:** After exiting the door, Alice should ~~turn left to go down the stairs~~ and cross the street using the pedestrian crossing. This is because the stairs leading down and the pedestrian crossing are located to the left of the building entrance, as indicated by the position of the yellow fire hydrant (label 1) and the bicycle rack (label 2) in the images. Turning left will lead her towards these features and ultimately to the street where she can use the pedestrian crossing.*
>
> ***LongVILA:** After exiting the door, Alice should ~~turn left to head~~ towards the stairs and the pedestrian crossing. This is indicated by the position of the door in image 0, where the stairs and the pedestrian crossing are located to the left of the door when facing outwards. The red circles labeled '1' and '10' in the images show the direction of the stairs and the pedestrian crossing, respectively, which are both to the left of the door.*
>
> For the 3D understanding tasks, Oryx demonstrates a stronger ability for spatial imagination compared to our baselines. In the challenging task involving direction in a first-person view, both LLaVA-OneVision and LongVILA provide incorrect conclusions. In contrast, Oryx accurately advises turning right, supported by the correct reasoning process.
>
> ---
>
> **References**
>
> [1] Bo Li, et al. LLaVA-OneVision: Easy Visual Task Transfer
>
> [2] Fuzhao Xue, et al. Longvila: Scaling long-context visual language models for long videos

---

> ### Author Response · Authors · 2024-11-25
> **Official Comment by Authors**
>
> Dear Reviewer ceNe:
>
> We sincerely appreciate your great efforts in reviewing this paper.   Your constructive advice and insightful comments really help improve our paper.   We have carefully tried to address your concerns with detailed explanations, essential experiments, and the revised version of our manuscripts.
>
> Considering the approaching deadline, we would appreciate it if you could spare some time to go over our response, and we can further address unclear explanations and remaining concerns if any.   We sincerely hope that you can consider reevaluating the initial assessment if we have successfully addressed your concerns.
>
> Best Regards,
>
> The Authors

---

### Comment · Area_Chair_fCuS · 2024-11-24

Dear Reviewers,

This is a friendly reminder that the discussion period will end on Nov 26th (Anywhere on Earth). If you have not already, please take a careful look at the other reviews and author responses, and comment on whether your original rating stands. Thank you.

Best, AC

---

### Author Response · Authors · 2024-11-25
**General Response by Authors**

We would like to express our gratitude to the reviewers for their positive feedback and insightful suggestions. We have uploaded a revised version of our paper and highlighted the modifications in blue color. The modifications are concluded below:

1. We optimize the flow in the method part and simplify the technical details to make the method more concentrated. We adjust the architecture and the data curation part to make the logic more clear. (Reviewer hrJh)

2. We add an apple-to-apple comparison with state-of-the-art MLLM in Appendix A. (Reviewer ceNe)



3. We add the failure cases and the relevant analysis in Appendix B. (Reviewer hrJh)



4. We test the inference speed, memory efficiency, and the memory-resolution curve to strongly demonstrate the efficiency of OryxViT in Appendix C.2 (Reviewer ceNe, hrJh)



5. We add an ablation about the proposed training data about long-form videos and 3D data in Appendix C.3 (Reviewer hrJh)



6. We evaluate the design of the MLP adapter in Appendix C.4 (Reviewer hrJh)



7. We add a detailed analysis on the downsampling strategy and function in Appendix C.5 (Reviewer hrJh, qcfr)



8. We provide details about training OryxViT in Appendix D.2 (Reviewer o36J)



We hope the revised version can better address your concern and we sincerely thank your effort in reviewing the paper.

---

### Meta-Review · Area_Chair_fCuS · 2024-12-19

**Metareview:**

This paper presents the Oryx series, a unified multi-modal framework for spatio-temporal understanding that handles diverse visual inputs using OryxViT for native resolution processing, a Dynamic Compressor for efficient data handling, and a joint training strategy.  The initial scores were 5,5,6,6.  Strengths included innovative approach that supports native input resolution, extensive experiments, and promising results.  Weaknesses included incremental improvements, some overclaiming statements, and some missing comparisons and experimental analysis.  The rebuttal and discussion largely addressed these concerns, and two reviewers increased their scores, leading to a final score of 6,6,6,6.  The AC agrees with the unanimous decision of the reviewers to accept the paper.

**Additional Comments On Reviewer Discussion:**

Strengths included innovative approach that supports native input resolution, extensive experiments, promising results.  Weaknesses included incremental improvements, some overclaiming statements, and some missing comparisons and experimental analysis.  The rebuttal and discussion largely addressed these concerns, and two reviewers increased their scores, leading to a final score of 6,6,6,6.  The AC agrees with the unanimous decision of the reviewers to accept the paper.

---

### Decision · Program_Chairs · 2025-01-22

Accept (Poster)